# Loss of immunity-related GTPase GM4951 leads to nonalcoholic fatty liver disease without obesity

Zhao Zhang 1,2✉, Yu Xun1,2, Shunxing Rong3,4, Lijuan Yan1, Jeffrey A. SoRelle 1, Xiaohong Li1, Miao Tang1, Katie Keller1, Sara Ludwig1, Eva Marie Y. Moresco 1 & Bruce Beutler 1✉

Obesity and diabetes are well known risk factors for nonalcoholic fatty liver disease (NAFLD), but the genetic factors contributing to the development of NAFLD remain poorly understood. Here we describe two semi-dominant allelic missense mutations (*Oily* and *Carboniferous*) of *Predicted gene 4951* (*Gm4951*) identified from a forward genetic screen in mice. GM4951 deficient mice developed NAFLD on high fat diet (HFD) with no changes in body weight or glucose metabolism. Moreover, HFD caused a reduction in the level of *Gm4951*, which in turn promoted the development of NAFLD. Predominantly expressed in hepatocytes, GM4951 was verified as an interferon inducible GTPase. The NAFLD in *Gm4951* knockout mice was associated with decreased lipid oxidation in the liver and no defect in hepatic lipid secretion. After lipid loading, hepatocyte GM4951 translocated to lipid droplets (LDs), bringing with it hydroxysteroid 17β-dehydrogenase 13 (HSD17B13), which in the absence of GM4951 did not undergo this translocation. We identified a rare non-obese mouse model of NAFLD caused by GM4951 deficiency and define a critical role for GTPase-mediated translocation in hepatic lipid metabolism.

[1] Center for the Genetics of Host Defense, University of Texas Southwestern Medical Center, Dallas, TX 75390, USA. [2] Division of Endocrinology, Department of Internal Medicine, University of Texas Southwestern Medical Center, Dallas, TX 75390, USA. [3] Center for Human Nutrition, University of Texas Southwestern Medical Center, Dallas, TX 75390, USA. [4] Department of Molecular Genetics, University of Texas Southwestern Medical Center, Dallas, TX 75390, USA. ✉email: Zhao.Zhang@UTSouthwestern.edu; Bruce.Beutler@UTSouthwestern.edu

Nonalcoholic fatty liver disease (NAFLD) is a condition in which excess fat is stored in the liver in people who drink little or no alcohol. NAFLD has become the most common cause of liver disease worldwide with an estimated global prevalence of 25.24%[1]. Isolated hepatosteatosis is the most frequent and least severe form of NAFLD, which also encompasses the more serious nonalcoholic steatohepatitis (NASH), cirrhosis, and hepatocellular carcinoma[2–4]. The development of NAFLD is a complex process determined by both environmental cofactors and genetic predisposition, and is strongly associated with obesity, insulin resistance, and type 2 diabetes[5]. In recent decades, it has been suggested that lifestyle changes drive the prevalence of NAFLD. However, hepatic fat content varies substantially among individuals with equivalent adiposity, indicating that genetic factors contribute to NAFLD.

We have performed a screen for NAFLD in mice with randomly induced germline mutations, sensitized by a high fat diet (HFD). Screening was coupled to automated meiotic mapping (AMM), which permits identification of causative mutations in real time[6]. Two semi-dominant allelic missense mutations (*Oily* and *Carboniferous*) of *Predicted gene 4951* (*Gm4951*) were discovered using this approach. As distinct from most mouse NAFLD models, which are associated with obesity, GM4951 deficient mice developed hepatosteatosis with no change in body weight. *Gm4951* is one of 20 "immunity-related GTPase" (IRG) genes in the mouse, which are organized in tandem gene clusters on chromosomes 11 and 18[7,8]. The encoded proteins, including GM4951, contain a GTP-binding domain with sequence homology to that of other GTPases[9]. Members of the IRG gene family are transcriptionally induced by interferons and are thought to be important for resistance to intracellular pathogens including *T. gondii*, *M. tuberculosis*, *S. typhimurium*, and *C. trachomatis*[7,8].

One hallmark of hepatosteatosis is the accumulation of lipid droplets (LDs), ubiquitous organelles important for lipid storage and consequently, for metabolism[10]. LD cores contain neutral lipids and can grow and shrink in response to different signals[11]. LD surfaces are decorated by specific proteins that are the major executors of LD functions[12]. Hydroxysteroid 17β-dehydrogenase 13 (HSD17B13) is a LD-associated protein expressed specifically in the liver[13,14]. Recently, several reports have linked variants of HSD17B13 with the development of NAFLD both in mice and humans[15–18]. The molecular mechanism(s) underlying NAFLD, and identities of HSD17B13-associated proteins remain largely uncharacterized.

In this work we demonstrate that GM4951 interacts with HSD17B13, transporting HSD17B13 to LDs in a GTPase-dependent manner to regulate hepatic lipid metabolism.

## Results

**Identification of *Oily* and *Carboniferous* alleles.** Wild type (WT) C57BL/6 J mice fed a HFD for four weeks develop NAFLD, as detected by elevated liver triglyderides relative to levels prior to HFD feeding. We screened for mutations that modify the NAFLD phenotype and identified a phenotype termed *Oily*, characterized by increased liver triglycerides compared to WT mice after four weeks on a HFD (Fig. 1a). The *Oily* phenotype was mapped as a quantitative trait. Liver triglycerides were measured in a total of 44 G3 mice and normalized to total protein. AMM[6] implicated a missense allele of *Gm4951* as the causative mutation, displaying strongest linkage in an additive model of inheritance ($P = 1.66 \times 10^{-5}$) (Fig. 1b). The *Oily* mutation was a single nucleotide transversion from C to A, causing substitution of a lysine for an asparagine at position 86 (N86K) in the GM4951 protein (Supplementary Fig. 1a). Later, a second allele of *Gm4951*, termed *Carboniferous*, was also identified by our screen (Fig. 1c).

*Carboniferous* also scored in an additive model of inheritance ($P = 2.332 \times 10^{-5}$) (Fig. 1d). The *Carboniferous* mutation was a single nucleotide transition from A to G, causing substitution of a glycine for an aspartic acid at position 125 (D125G) in the GM4951 protein (Fig. S1a). Combination of these two pedigrees in superpedigree analysis[6] revealed a significant *P* value of $5.16 \times 10^{-10}$ in an additive model of inheritance (Fig. 1e, f). Body weight before and after HFD, fasting and refeeding glucose, fasting and refeeding insulin, and fasting and refeeding free fatty acids (FFA) were also measured during the course of screening (Supplementary Fig. 2). None of these phenotypes were significantly different in reference allele homozygotes as compared to homozygous or heterozygous *Oily* mice (Supplementary Fig. 2). These data suggested that *Oily* and *Carboniferous* mutations caused a form of NAFLD unassociated with obesity or abnormal glucose metabolism.

**Oily and Carboniferous are caused by loss-of-function mutations in *Gm4951*.** Neither the *Oily* (N86K) nor the *Carboniferous* (D125G) mutations affected stability of GM4951 as revealed by a similar expression level as the WT protein in 293T cells (Fig. 2a, b). We suspected that these mutations affect the function of GM4951 protein. By CRISPR/Cas9 gene targeting, we introduced a null allele of *Gm4951*, encoding the first 35aa of the GM4951 protein followed by six aberrant amino acids and a termination codon, into the germline of C57BL/6 J mice (Fig. 2a, Supplementary Fig. 1a). *Gm4951* knockout homozygotes appeared indistinguishable from WT mice and both sexes were fertile. Crosses of knockout heterozygotes yielded the anticipated Mendelian ratio among offspring at weaning age (Supplementary Fig. 1b). *Gm4951* knockout mice were bred and expanded to form a large pedigree with all three zygosities represented, and subjected to the same screens as G3 mice. The *Gm4951* null allele recapitulated all aspects of the *Oily* and *Carboniferous* phenotypes (Fig. 2c–l) and reproduced the additive inheritance pattern (Fig. 2c, d). These data confirm that *Gm4951* mutations in *Oily* and *Carboniferous* strains were causative of the observed phenotypes, and ruled out gain-of-function as a mechanism.

**The development of NAFLD in *Gm4951* knockout mice is not associated with alterations in body weight or glucose metabolism.** Obesity is strongly associated with an increased risk of NAFLD[19]. The presence of hepatosteatosis is often associated with aberrant glucose, fatty acid, and lipoprotein metabolism, reflective of insulin resistance and other metabolic dysfunctions[20,21]. It is still not clear whether NAFLD causes metabolic dysfunction or whether metabolic dysfunction is responsible for the development of NAFLD, or both[22,23]. Although obesity correlates with NAFLD, a minority of patients with NAFLD are lean[24]. Similarly, *Oily* mice and $Gm4951^{-/-}$ mice on a HFD did not show significant alterations in body weight or glucose metabolism concomitant with development of hepatosteatosis (Fig. 2, Supplementary Fig. 2). The inguinal white adipose tissue (iWAT) of 4 week HFD-fed $Gm4951^{-/-}$ mice expressed similar mRNA levels of lipid metabolism genes (*Acaca*, *Pnpla2*, *Dgat1*, *Fasn*), adipokines (*Fabp4*, *Adipoq*, *Leptin*), and key transcription factors (*Fabp4*, *Cebpa*) (Supplementary Fig. 3a). Besides parameters tested in our screen, we checked additional lipid profiles in serum and liver from 4 week HFD-fed mice. Compared with WT mice, $Gm4951^{-/-}$ mice exhibited increased serum triglycerides (Supplementary Fig. 3b) with no change in serum total cholesterol (Supplementary Fig. 3c), low-density lipoprotein (LDL) cholesterol (Supplementary Fig. 3d), and high-density lipoprotein (HDL) cholesterol (Supplementary Fig. 3e). GM4951 deficiency significantly increased both triglycerides

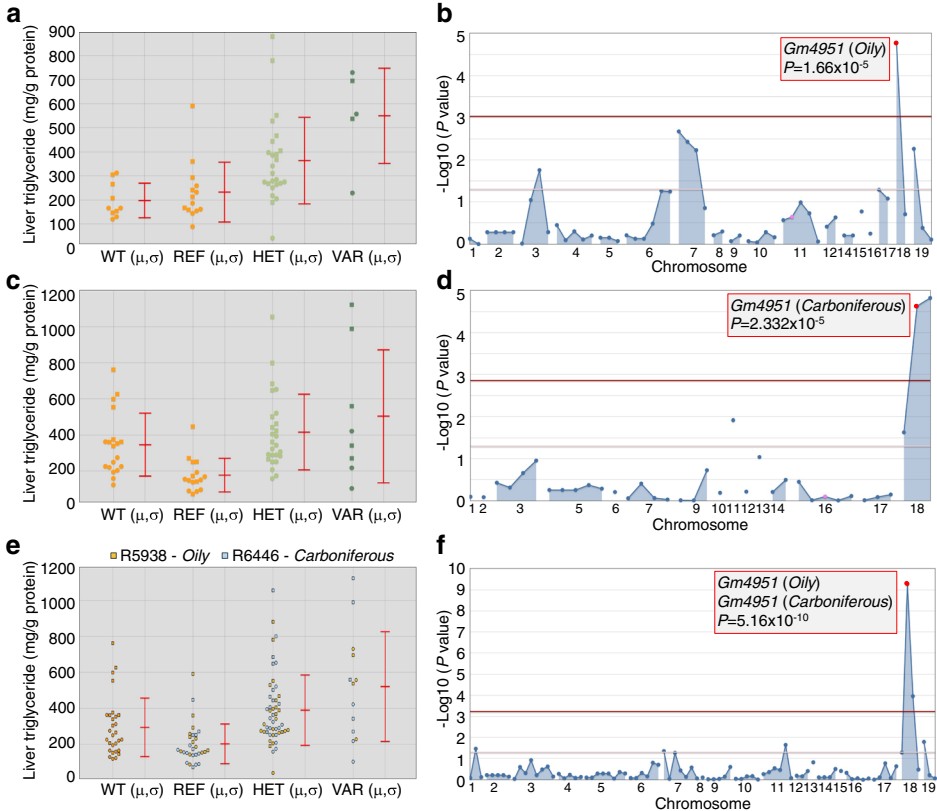

**Fig. 1 Identification and mapping of *Oily* and *Carboniferous*. a** Liver triglyceride data from G3 mice plotted vs. genotype at the *Oily* mutation site of *Gm4951* after four weeks on a HFD ($n = 10$ mice in WT; $n = 14$ mice in REF; $n = 25$ mice in HET; $n = 5$ mice in VAR). **b** Manhattan plot showing P values calculated using a likehood-ratio test from an additive model of inheritance. **c** Liver triglyceride data from G3 mice plotted vs. genotype at the *Carboniferous* mutation site of *Gm4951* after four weeks on a HFD ($n = 19$ mice in WT; $n = 16$ mice in REF; $n = 26$ mice in HET; $n = 8$ mice in VAR). **d** Manhattan plot showing P values calculated using a likehood-ratio test from an additive model of inheritance. **e, f** Liver triglyceride data from pedigrees R5938 (*Oily*) and R6446 (*Carboniferous*) were pooled together in superpedigree analysis and plotted as in **a** and **c** ($n = 29$ mice in WT; $n = 30$ mice in REF; $n = 51$ mice in HET; $n = 13$ mice in VAR); Manhattan plot as in (**b**) and (**d**). P values were calculated using a likehood-ratio test from an additive model of inheritance. WT, C57BL/6 J mice age-matched with G3 mice; REF, G3 mice homozygous for the reference allele of the indicated gene; HET, G3 mice heterozygous for the reference allele and for the mutant allele; VAR, G3 mice homozygous for the mutant allele. Each data point represents one mouse; mean ($\mu$) and SD ($\sigma$) are indicated (**a**, **c**, **e**). Data are from one experiment.

(Supplementary Fig. 3f) and FFA (Supplementary Fig. 3g) in the liver, with no change in liver cholesterol (Supplementary Fig. 3h). To check if *Gm4951*$^{-/-}$ mice develop NAFLD on a standard chow diet, we monitored liver triglycerides at different ages in WT and *Gm4951*$^{-/-}$ mice (Fig. 3a). Significantly increased liver triglycerides were observed in *Gm4951* knockout mice by 24 weeks of age (Fig. 3a). Along with the development of hepatosteatosis, we also detected an elevated level of alanine transaminase (ALT) in the serum of *Gm4951*$^{-/-}$ mice, indicating liver damage caused by hepatosteatosis (Fig. 3b). On a chow diet, GM4951 deficiency changed neither the body weight nor body fat content measured by magnetic resonance imaging (MRI) (Fig. 3c, d). Also, fasting glucose, insulin, and serum triglycerides remained the same in *Gm4951*$^{-/-}$ and WT mice (Fig. 3e–g). These data suggest that NAFLD develops spontaneously in GM4951 deficient mice with no noticeable obesity or insulin resistance.

In screening, a 4-week HFD was used to augment the development of NAFLD in C57BL6/J mice. To maximize the obesogenic effect of HFD, *Gm4951*$^{-/-}$ mice were fed a HFD for 24 weeks. Compared to the 4-week course of HFD, a long-term HFD further increased liver weight (Fig. 3h), liver triglycerides (Fig. 3i), and serum ALT in *Gm4951*$^{-/-}$ mice (Fig. 3j). Despite the further development of NAFLD, no significant differences in body weight, fat weight, glucose, insulin, and serum triglycerides were observed between long-term HFD-fed WT

and *Gm4951*$^{-/-}$ mice (Fig. 3k–o). Histologically, compared to *Gm4951*$^{+/+}$ controls, *Gm4951*$^{-/-}$ mice fed a long-term HFD exhibited severe hepatosteatosis as detected by both hematoxylin and eosin (H&E) staining (Fig. 3p, q, Supplementary Fig. 4a) and oil red O staining (Fig. 3r, s, Supplementary Fig. 4b) of liver sections. Long-term HFD fed WT and *Gm4951*$^{-/-}$ mice were also checked for different aspects of liver fibrosis which is a prognostic factor for NAFLD patients[25]. Compared with WT mice, *Gm4951*$^{-/-}$ mice had increased collagen in the liver detected by picrosirius red (PSR) staining (Fig. 3t, u, Supplementary Fig. 4c). We also observed increased α-smooth muscle actin (α-SMA)-positive myofibroblasts in *Gm4951*$^{-/-}$ liver sections, suggesting the activation of hepatic stellate cells (HSCs) (Fig. 3v, w, Supplementary Fig. 4d). The overall NAFLD Activity Score (NAS)[26] was increased for livers from *Gm4951*$^{-/-}$ mice (Supplementary Fig. 4e), suggesting the development of NASH in *Gm4951*$^{-/-}$ mice after long-term HFD. Along with this, the mRNA levels of several liver fibrosis-related genes (*Col1a1*, *Mmp9*, *Timp1*) were also increased in livers from *Gm4951*$^{-/-}$ mice (Fig. 3x). To determine the effect of GM4951 overexpression on liver triglycerides, we used hydrodynamic tail vein injection to deliver 3xFlag-Gm4951 expression plasmid to the liver. Two weeks after HFD, 3xFlag-Gm4951-injected mice had a decreased level of liver triglycerides compared with control vector-injected mice (Fig. 3y, z). All

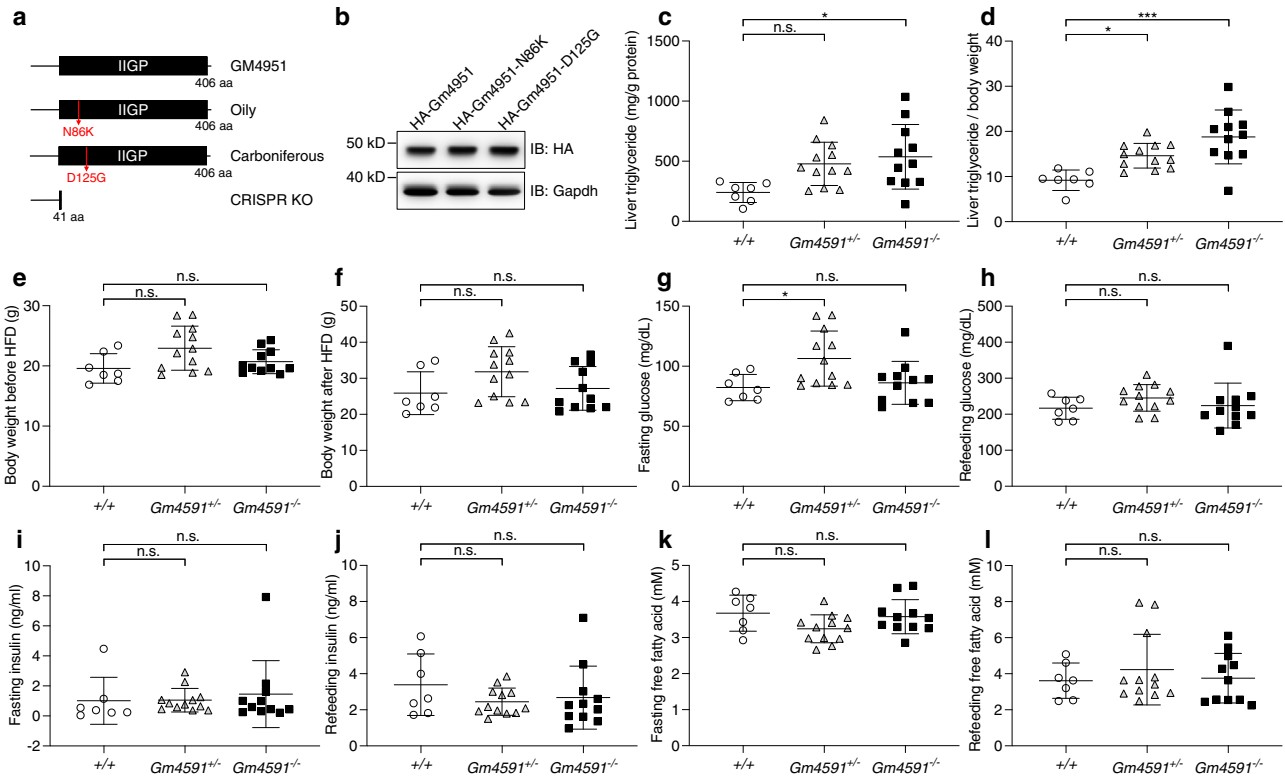

**Fig. 2 Mutations in *Gm4951* cause the *Oily* and *Carboniferous* phenotypes. a** Protein domain diagrams of mouse GM4951 WT form, with the *Oily* mutation, *Carboniferous* mutation, and truncated form generated by CRISPR KO. **b** Immunoblots of lysates of 293T cells expressing HA-tagged WT GM4951, GM4951-N86K, and GM4951-D125G. **c–l** Liver triglycerides (**c**), liver triglycerides normalized by body weight (**d**), body weight before HFD (**e**), body weight after four weeks of HFD (**f**), overnight fasting glucose (**g**), glucose 2 h after refeeding (**h**), overnight fasting insulin (**i**), insulin 2 h after refeeding (**j**), overnight fasting free fatty acid (**k**), and free fatty acid 2 h after refeeding (**l**) plotted for WT (+/+, $n = 7$), heterozygous (*Gm4951*[+/−], $n = 12$), and homozygous *Gm4951* CRISPR KO (*Gm4951*[−/−], $n = 11$) mice. Data in (**c**, **d**, and **f**) were measured after 4 weeks on a HFD; data in (**g–l**) were measured after two weeks on a HFD. Each data point represents one mouse (**c–l**). Data are presented as means ± SD. *P* values were determined by one-way ANOVA with Tukey's multiple comparison test. *P* values are denoted by *$P < 0.05$; ***$P < 0.001$; ns, not significant with $P > 0.05$. The exact *P* values of statistically significant groups are: 0.0149 (**c**, +/+ vs *Gm4951*[−/−]); 0.0283 (**d**, +/+ vs *Gm4951*[+/−]); 0.0002 (**d**, +/+ vs *Gm4951*[−/−]); 0.0323 (**g**, +/+ vs *Gm4951*[+/−]). Data are representative of two independent experiments (**b–l**).

these data suggest a liver intrinsic role of GM4951 in the regulation of liver triglycerides.

**GM4951 is an interferon inducible GTPase highly expressed in hepatocytes**. To characterize the function of GM4951, we first studied the expression profile of *Gm4951* mRNA in different mouse tissues. Among tissues tested from 8-week-old male C57BL6/J mice, *Gm4951* mRNA was maximally expressed in the liver (Fig. 4a). *Gm4951* mRNA was also detected in the inter-scapular white adipose tissue (iWAT), lung, epididymal white adipose tissue (eWAT), stomach, and spleen, but in each case, at levels far lower than in liver (Fig. 4a). To measure relative expression of the endogenous GM4951 protein, we generated 3xFlag-*Gm4951* knockin mice by CRISPR-mediated homologous replacement (Fig. 4b). Consistent with the *Gm4951* mRNA profile, endogenous GM4951 protein (as indicated by anti-Flag tag immunoblotting) showed highest expression level in the liver, with detectable amounts in the spleen, lung, eWAT, iWAT, thymus, and intestine (Fig. 4c). 90% of liver mass is composed of hepatocytes, while the remainder consists of nonparenchymal elements including Kupffer cells, liver sinusoidal endothelial cells, hepatic stellate cells, and cholangiocytes. Previously reported mass spectrometry-based protein quantification revealed highest expression of GM4951 protein in hepatocytes[27] (Fig. 4d). The subcellular localization of GM4951 in AML12 mouse hepatocytes appeared perinuclear and partially colocalized with markers for

mitochondria (apoptosis-inducing factor, AIF), endosomes (early endosome antigen 1, EEA1), ER (protein disulfide isomerase, PDI), and Golgi (receptor-binding cancer antigen expressed on SiSo cells, RCAS1) (Fig. 4e–h).

GM4951 contains an interferon inducible GTPase (IIGP) domain, which is a Pfam[28] domain (#PF05049) built on 6 seed sequences of IRG protein family (TGTP1, IRGA6, IFI47, IRGM1, IGTP, IRGM2) for alignment. Considering the predominant expression of GM4951 in hepatocytes, we treated mouse primary hepatocytes with recombinant IFNγ in vitro. Induction of *Gm4951* mRNA occurred in response to IFNγ administered at a concentration between 0.1 and 1.0 ng/ml (Fig. 4i). Consistent with the induction of *Gm4951* mRNA, we also detected increased expression of GM4951 protein in hepatocytes isolated from 3xFlag-*Gm4951* knockin mice and stimulated with IFNγ in vitro (Fig. 4j). To test whether GM4951 is a bona fide GTPase, recombinant full-length GM4951 protein was expressed in *E. coli BL21* and purified with affinity chromatography and size-exclusion chromatography. During the bioluminescent detection of GTPase activity, purified GM4951 protein exhibited concentration-dependent intrinsic GTPase activity (Fig. 4k). These results suggest that GM4951 harbors common features of IRG family proteins.

Among 20 IRG proteins in mice, GM4951 displays the highest sequence identity (77%) with interferon inducible GTPase 1 (IIGP1, now known as IRGA6) (Supplementary Fig. 5a). The

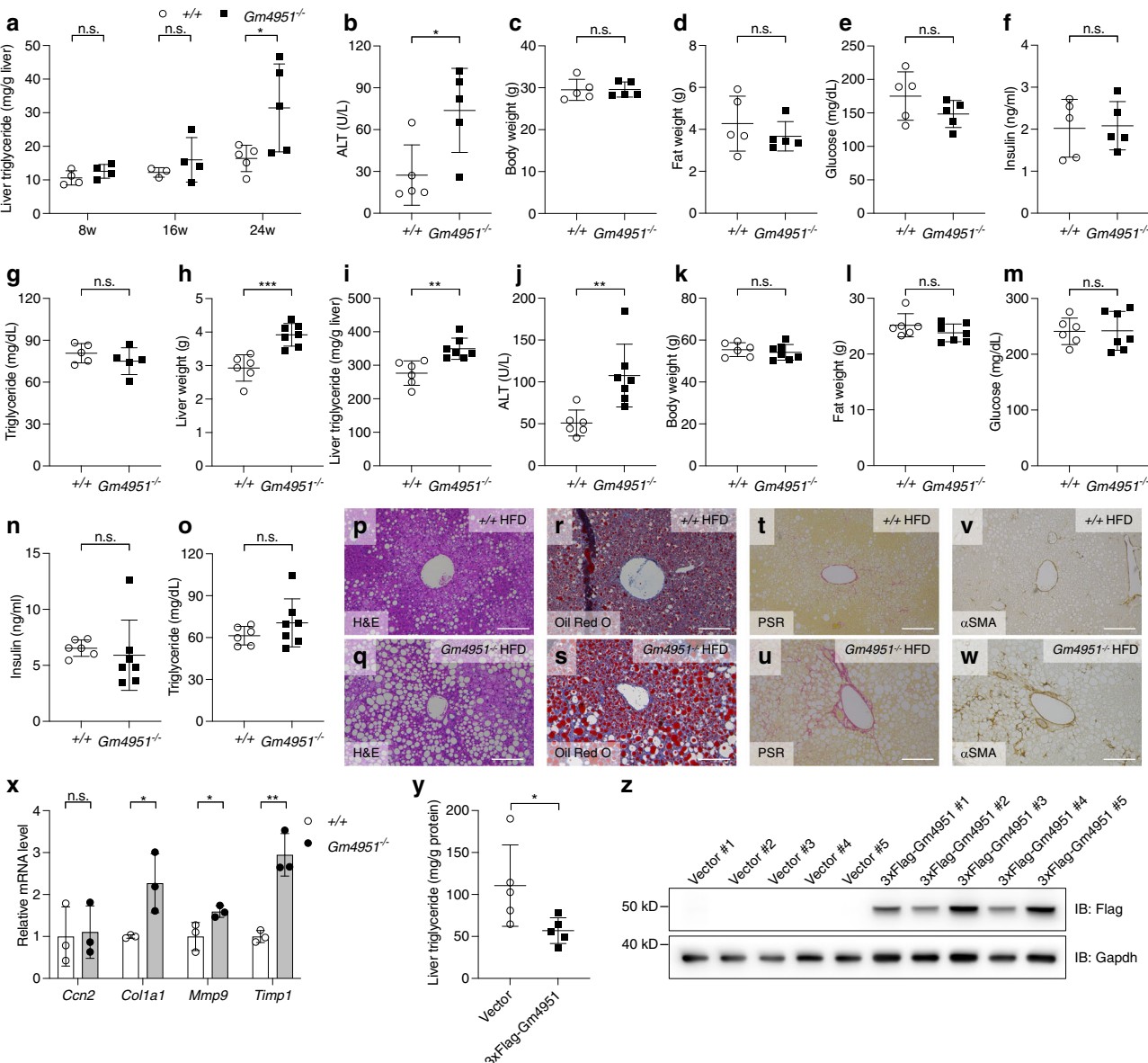

**Fig. 3 The development of NAFLD in *Gm4951*−/− mice on chow diet and HFD. a** Liver triglycerides in WT and *Gm4951*−/− mice on chow diet for eight weeks (*n* = 4 in WT; *n* = 4 in *Gm4951*−/−), 16 weeks (*n* = 3 in WT; *n* = 4 in *Gm4951*−/−), and 24 weeks (*n* = 5 in WT; *n* = 5 in *Gm4951*−/−). **b–g** Serum ALT (**b**), body weight (**c**), fat weight (**d**), fasting glucose (**e**), fasting insulin (**f**), and fasting serum triglycerides (**g**) of WT (*n* = 5) and *Gm4951*−/− (*n* = 5) mice on chow diet for 24 weeks. **h–o** Liver weight (**h**), liver triglycerides (**i**), serum ALT (**j**), body weight (**k**), fat weight (**l**), fasting glucose (**m**), fasting insulin (**n**), and fasting serum triglycerides (**o**) of WT (*n* = 6) and *Gm4951*−/− (*n* = 7) mice on HFD for 24 weeks. **p–w** H&E staining (**p** and **q**), Oil Red O staining (**r** and **s**), PSR staining (**t** and **u**), and αSMA staining (**v** and **w**) of liver sections from WT and *Gm4951*−/− mice on HFD for 24 weeks. (Scale bars: 100 μm). **x** Relative mRNA level of different fibrosis-related genes in the liver from WT and *Gm4951*−/− mice on HFD for 24 weeks (*n* = 3 mice per genotype). Levels were normalized to *Polr2a* mRNA and then to levels in WT mice. **y** Overexpression of 3xFlag-Gm4951 by hydrodynamic tail vein injection. Liver triglycerides in mice injected with control vector or 3xFlag-Gm4951 vector and fed HFD for two weeks (*n* = 5 mice in each group). **z** Immunoblot analysis of 3xFlag-Gm4951 protein expression in mouse livers two weeks after hydrodynamic tail vein injection. Each data point represents one mouse. Data are presented as means ± SD. *P* values were determined by two-tailed Student's *t* test. *P* values are denoted by \**P* < 0.05; \*\**P* < 0.01; \*\*\**P* < 0.001; ns, not significant with *P* > 0.05. The exact *P* values of statistically significant groups are: 0.0177 (**a**, 24w); 0.0234 (**b**); 0.0005 (**h**); 0.0026 (**i**); 0.0055 (**j**); 0.0355 (**x**, *Col1a1*); 0.0466 (**x**, *Mmp9*); 0.0031 (**x**, *Timp1*); 0.0463 (**y**). Data are representative of two independent experiments (**a–o**, **x–z**) or from one experiment (**p–w**).

crystal structure of IRGA6 revealed that GTP-dependent oligomerization of IRGA6 is required for cooperative GTP hydrolysis[9]. We constructed both the *Oily* mutant form (N86K) and *Carboniferous* mutant form (D125G) of GM4951, each tagged with both HA and 3xFlag, and tested whether either mutation might disrupt dimerization. Overexpressed WT GM4951 protein in 293T cells interacted strongly with itself

(Supplementary Fig. 5b). Neither *Oily* nor *Carboniferous* mutations affected self-interaction. As expected, the predicted dimerization interface mutation (E43A-K47A) strongly disrupted the self-interaction of GM4951 in 293T cells (Supplementary Fig. 5c).

Structural study of IRGA6 revealed active site residues critical for GDP and GppNHp binding, including G81/K82 (P loop/G1),

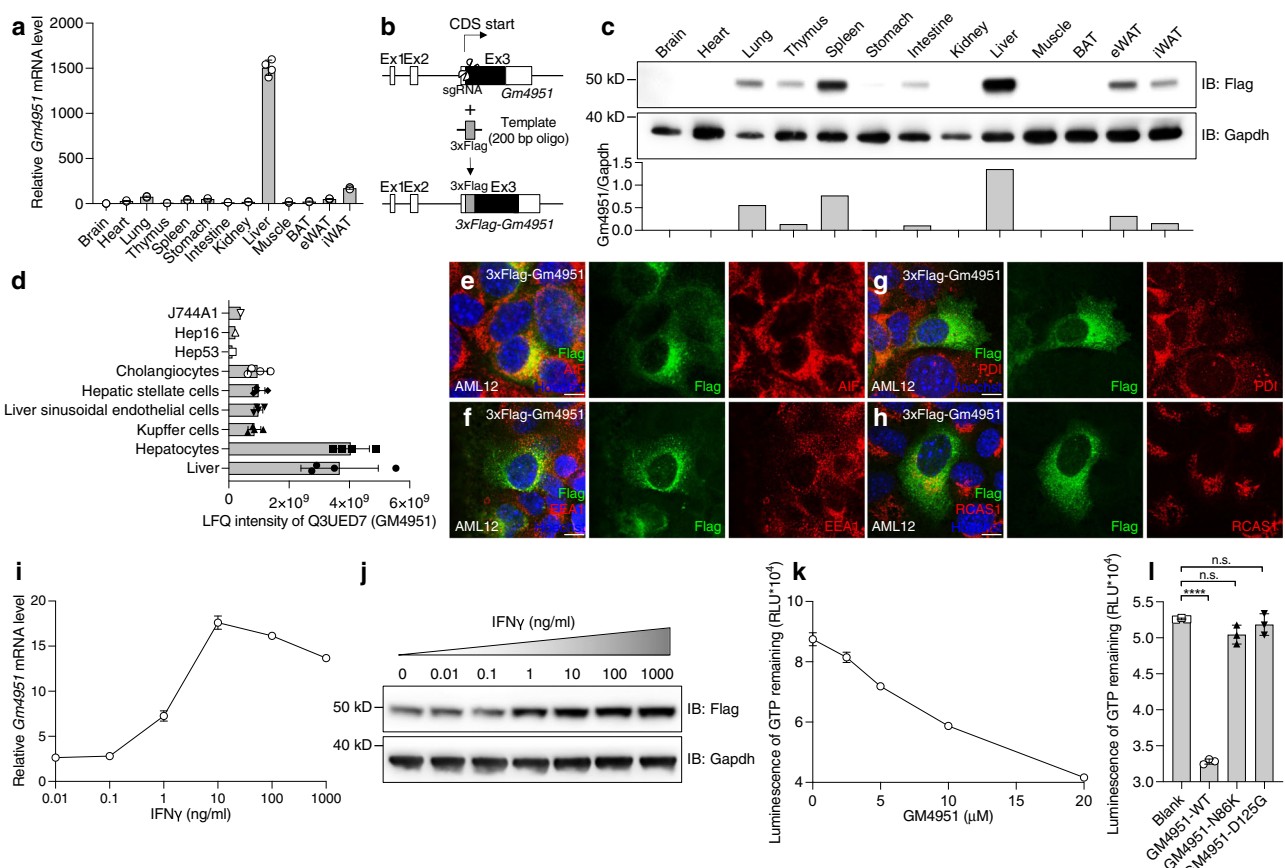

**Fig. 4 GM4951 is an interferon inducible GTPase that is highly expressed in hepatocytes. a** Relative *Gm4951* mRNA level in different mouse tissues normalized by *Polr2a* (*n* = 3 mice). **b** Generation of 3xFlag tagged *Gm4951* knockin mice by CRISPR. **c** Immunoblot analysis of 3xFlag-Gm4951 protein expression in different mouse tissues (eight weeks old male). Gapdh was used as a loading control. Quantification of bands in the lower panel are shown as ratio of intensity of Gm4951 to Gapdh. **d** Label-free quantification (LFQ) intensity data[27] for GM4951 protein in different primary cells isolated from the mouse liver or cell lines (*n* = 1 assay for J744A1, Hep16, and Hep53; *n* = 4 assays for other cells). **e–h** AML12 cells expressing 3xFlag-tagged Gm4951 were immunostained with Flag antibody (green), Hoechst 33342 (blue) to visualize nuclei, and organelle markers (red): AIF (mitochondria), EEA1 (endosome), PDI (ER), and RCAS1 (Golgi apparatus). (Scale bar: 10 μm.) **i** Relative mRNA level of *Gm4951* in primary hepatocytes treated with different doses of IFNγ for 24 h (*n* = 3 independent cultures). **j** Immunoblots of lysates of primary hepatocytes isolated from *3xFlag-Gm4951* knockin mice. Different doses of IFNγ were added into the medium for 24 h. Gapdh was used as a loading control. **k** Luminescence to quantitate the amount of GTP remaining 1 h after the GTPase assay with different concentrations of purified GM4951 protein (*n* = 3 assays per condition). **l** Luminescence to quantitate the amount of GTP remaining 1 h after the GTPase assay with purified GM4951 WT and mutated proteins (*n* = 3 assays per condition). Each data point represents one mouse (**a**) or one reaction (**d**, **l**). Data are presented as means ± SD. *P* values were determined by one-way ANOVA with Tukey's multiple comparison test. *P* values are denoted by ****\*P* < 0.0001; ns, not significant with *P* > 0.05. The exact *P* value of statistically significant group is 3.473 × 10⁻⁸ (Blank vs GM4951-WT). Data are representative of two independent experiments (**a**, **c**, **e–l**).

D126 (126DLPG129/G3), D186 (183TKVD186/G4), and S231/N232 (G5)[9]. All these amino acids are identical in GM4951. The D125 in GM4951 is equivalent to D126 in IRGA6, which is in close contact with Mg²⁺ as shown in the IRGA6-GDP structure (Supplementary Fig. 5d). The N86 in GM4951 is equivalent to N87 in IRGA6, which is adjacent to P loop (Supplementary Fig. 5e). Both purified *Oily* (N86K) and *Carboniferous* (D125G) mutated GM4951 showed dramatically reduced intrinsic GTPase activity compared with WT GM4951 protein in vitro (Fig. 4l). Taken together, these data indicate that GM4951 is a hepatocyte interferon-inducible protein with intrinsic GTPase activity. In addition, they suggest that the *Oily* and *Carboniferous* mutations disrupt the ability of GM4951 to hydrolyze GTP, without affecting its dimerization.

**GM4951 supports lipid oxidation.** Hepatic lipid accumulation results from an imbalance between lipid acquisition and lipid disposal, which are regulated through four major pathways:

uptake of circulating lipid, de novo lipogenesis, fatty acid oxidation, and export of lipids via very low-density lipoproteins (VLDL). To investigate changes in expression of representative genes encoding components of these four major pathways, a group of WT and *Gm4951⁻/⁻* mice was placed on HFD for four weeks while another age-matched group of WT and *Gm4951⁻/⁻* mice was placed on a standard chow diet. The mRNA levels of two lipid transporters (*Slc27a1*, *Cd36*), two lipid exporters (*Apob*, *Mttp*), and four lipogenesis enzymes (*Acaca*, *Fasn*, *Elovl6*, *Scd1*) were similar between WT and *Gm4951⁻/⁻* livers both in chow and HFD groups (Fig. 5a, b). Significantly decreased expression of *Ppara* and *Acox1* were observed in livers from *Gm4951⁻/⁻* mice on HFD, and a trend toward decreased expression (not significant) of *Acot1* and *Acadl* were observed in *Gm4951⁻/⁻* mice on HFD (Fig. 5b). These four genes encode constituents of lipid oxidation pathways. No significant changes of these lipid oxidation genes were observed in *Gm4951⁻/⁻* mice on chow diet at 10 weeks of age when NAFLD had not developed yet (Fig. 5a).

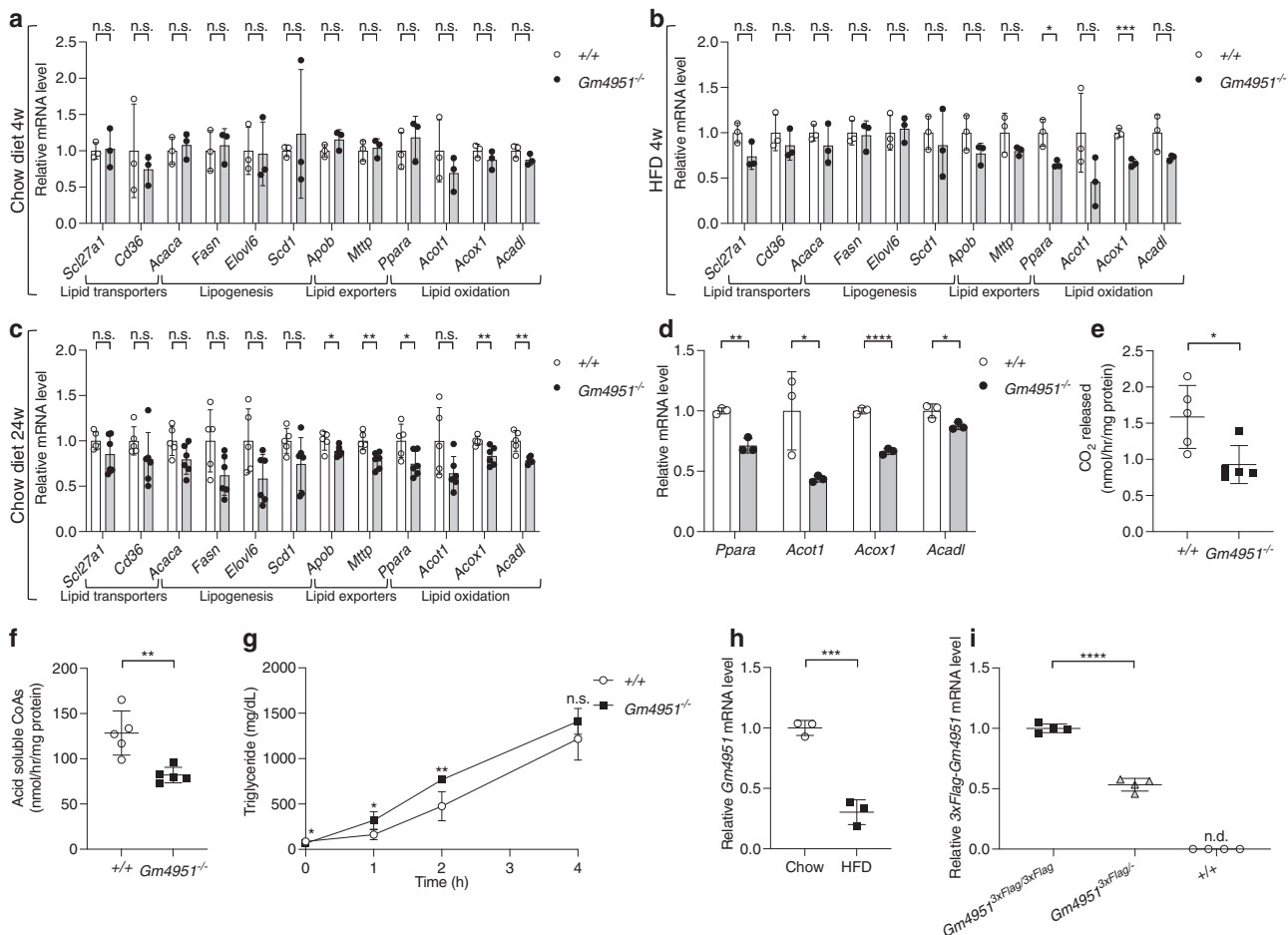

**Fig. 5 GM4951 is involved in lipid oxidation and is downregulated by HFD feeding. a–c** Relative mRNA level of different genes in the liver from WT and *Gm4951*⁻/⁻ mice on chow diet (**a**) and HFD (**b**) for four weeks (*n* = 3 mice per genotype), and chow diet for 24 weeks (**c**) (*n* = 5 mice per genotype). Levels were normalized to *Polr2a* mRNA and then to levels in WT mice. **d** Relative mRNA level of different genes in hepatocytes from WT and *Gm4951*⁻/⁻ mice after 0.5 mM OA for 24 h (*n* = 3 mice per genotype). **e**, **f** $CO_2$ released (**e**) and acid soluble CoAs (**f**) from fatty acid oxidation experiment were measured with scintillation counting and normalized with total protein of liver lysates (*n* = 5 mice per genotype). **g** The serum level of triglycerides at different time points after Triton WR-1399 injection in WT and *Gm4951*⁻/⁻ mice on HFD for 4 weeks (*n* = 3 mice per genotype). **h** Relative mRNA level of *Gm4951* in the liver from C57BL6/J mice on chow diet or HFD for 24 weeks (*n* = 3 mice per genotype). Levels were normalized to *Polr2a* mRNA and then to levels in WT mice fed chow diet. **i** Relative mRNA level of *3xFlag-Gm4951* in the liver from mice of the indicated genotypes on chow diet at eight weeks of age (*n* = 4 mice per genotype). n.d. not detected. Levels were normalized to *Polr2a* mRNA and then to levels in WT mice. Each data point represents one mouse (**a–f**, **h–i**). Data are presented as means ± SD. *P* values were determined by two-tailed Student's *t* test. *P* values are denoted by **P* < 0.05; ***P* < 0.01; ****P* < 0.001; *****P* < 0.001; ns, not significant with *P* > 0.05. The exact *P* values of statistically significant groups are: 0.0176 (**b**, *Ppara*); 0.0009 (**b**, *Acox1*); 0.0418 (**c**, *Apob*); 0.0041 (**c**, *Mttp*); 0.0281 (**c**, *Ppara*); 0.0017 (**d**, *Ppara*); 0.0401 (**d**, *Acot1*); 8.466 × 10⁻⁵ (**d**, *Acox1*); 0.0288 (**d**, *Acadl*); 0.0201 (**e**); 0.0039 (**f**); 0.0116 (**g**, 0 h); 0.0132 (**g**, 1 h); 0.0040 (**g**, 2 h); 0.0005 (**h**); 6.292 × 10⁻⁶ (**i**). Data are representative of two independent experiments (**a–d**, **g–i**) or from one experiment (**e–f**).

Reduced expression of these lipid oxidation genes was observed in livers from *Gm4951*⁻/⁻ mice on chow diet at 24 weeks of age when NAFLD had developed (Fig. 5c). Primary hepatocytes isolated from *Gm4951*⁻/⁻ mice also showed significantly decreased expression of the four lipid oxidation genes after oleic acid (OA) treatment, suggesting this phenotype is hepatocyte intrinsic (Fig. 5d). To assay lipid oxidation ability in the liver, we measured the reaction products from oxidation of ¹⁴C labeled palmitic acid by proteins in liver lysates from 4 week HFD fed mice in vitro. Compared with WT livers, *Gm4951*⁻/⁻ livers showed a significantly decreased rate of lipid oxidation as revealed by reduced $CO_2$ released and acid soluble CoAs in the reaction mix (Fig. 5e–f). To rule out a defect in triglyceride secretion, WT and *Gm4951*⁻/⁻ mice were put on HFD for 4 weeks which effectively induces NAFLD in *Gm4951*⁻/⁻ mice. After inhibition of lipoprotein lipase by Triton WR-1399 injection, we measured serum

triglycerides. We noticed a small but significantly higher level of triglycerides in the serum of *Gm4951*⁻/⁻ mice (Fig. 5g). This suggests that hepatosteatosis observed in *Gm4951*⁻/⁻ mice was not caused by failure to export triglycerides. We have shown that the development of NAFLD in GM4951 deficient mice was not associated with increased body weight or altered glucose metabolism (Fig. 3). Taken together, these data suggest that defective lipid oxidation pathways contribute to hepatic lipid accumulation in GM4951 deficient mice.

**HFD inhibits the expression of *Gm4951*.** The acceleration of NAFLD in GM4951 deficient mice by HFD (Fig. 3) prompted us to test whether the expression of *Gm4951* is regulated during the development of NAFLD. We fed C57BL6/J mice a chow diet or HFD for 24 weeks and checked the mRNA level of *Gm4951* in the

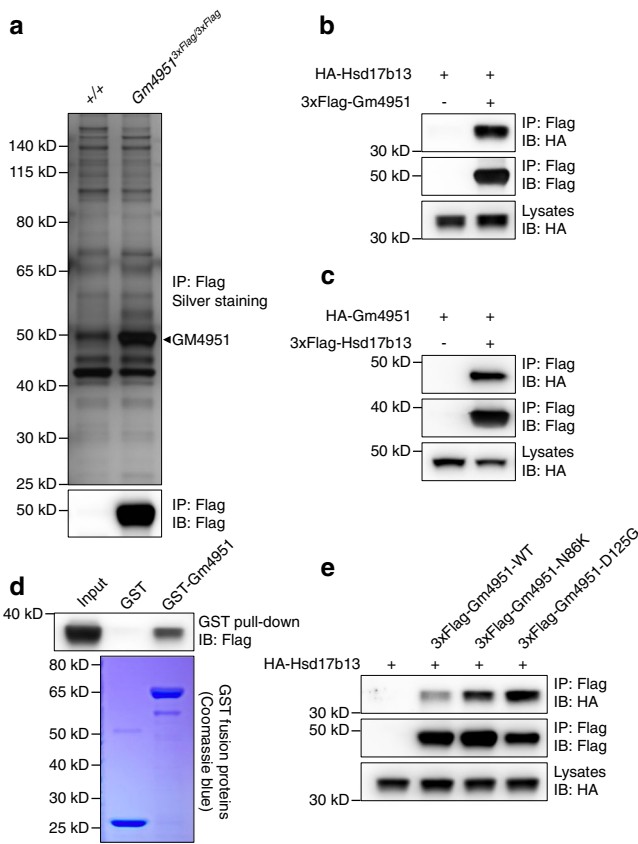

**Fig. 6 GM4951 interacts with HSD17B13. a** Silver staining (top) or immunoblot (bottom) of anti-Flag immunoprecipitates from +/+ or *Gm4951*<sup>3xFlag/3xFlag</sup> liver lysates. The same samples were analyzed by mass spectrometry to identify GM4951 interacting proteins. **b** Immunoblots of immunoprecipitates (top and middle) or lysates (bottom) of 293T cells expressing HA-tagged Hsd17b13 and 3xFlag-tagged Gm4951. **c** Immunoblots of immunoprecipitates (top and middle) or lysates (bottom) of 293T cells expressing HA-tagged Gm4951 and 3xFlag-tagged Hsd17b13. **d** Purified 3xFlag-tagged Hsd17b13 was incubated with GST or GST-tagged Gm4951. After GST pull-down, bound protein was analyzed by Flag immunoblot (top). The amounts of GST and GST-Gm4951 were visualized by Coomassie blue staining (bottom). **e** Immunoblots of immunoprecipitates (top and middle) or lysates (bottom) of 293T cells expressing HA-tagged Hsd17b13 and 3xFlag-tagged WT or mutant Gm4951. Data are representative of two independent experiments (**b**–**e**) or from one experiment (**a**).

liver by real-time PCR. Interestingly, the relative hepatic mRNA level of *Gm4951* in WT mice on HFD decreased to around 30% of that observed in mice on chow diet, suggesting a strong inhibition of hepatic *Gm4951* under HFD feeding (Fig. 5h). In mice fed a HFD for 4 weeks, an intermediate level of lipid accumulation was observed when one copy of *Gm4951* was mutated (*Gm4951*<sup>Oily/+</sup>, *Gm4951*<sup>Carboniferous/+</sup>) or deleted (*Gm4951*<sup>+/-</sup>) (Figs. 1, 2). We compared the mRNA level of *Gm4951* in mice with a single WT *Gm4951* allele with that in WT mice using 3xFlag-*Gm4951* knockin mice. Flag tag-specific real-time PCR primers were designed to amplify only 3xFlag-*Gm4951* but not WT *Gm4951*. mRNA expression derived from a single copy of *Gm4951* was approximately 53% of that derived from two copies of *Gm4951* in WT livers (Fig. 5i). These data suggest that the 70% reduction of *Gm4951* after long-term HFD feeding in WT mice may be a key step leading to hepatic steatosis. The downregulation of GM4951, caused by HFD, might in itself promote development of NAFLD.

**GM4951 interacts with HSD17B13.** To investigate the molecular mechanism of GM4951 in the regulation of lipid metabolism, we utilized our 3xFlag-*Gm4951* knockin mice to pull down GM4951 in liver lysates and identify interacting proteins by mass spectrometry (Fig. 6a). Among all proteins identified from mass spectrometry (Supplementary Data 1), HSD17B13, a recently identified liver restricted LD-associated protein, was considered highly relevant due to its association with human NAFLD[16,18]. GM4951 interacted with HSD17B13 when expressed in 293T cells (Fig. 6b, c). In vitro GST pull-down experiments supported a direct interaction between GM4951 and HSD17B13 (Fig. 6d). Interestingly, neither the *Oily* (N86K) nor *Carboniferous* (D125G) mutations disrupted the interaction of GM4951 with HSD17B13 (Fig. 6e). We checked the protein level of HSD17B13 in the liver of WT and *Gm4951*<sup>-/-</sup> mice, and no significant changes were observed when mice were fed with chow diet or HFD (Supplementary Fig. 6). These data suggest that the interaction between GM4951 and HSD17B13 is not dependent on the GTPase activity of GM4951 and does not regulate the protein level of HSD17B13 in the liver.

**GM4951 is necessary for HSD17B13 translocation to LDs.** To better understand the association of GM4951 and HSD17B13 in lipid metabolism, we checked their subcellular localization in hepatocytes. In untreated mouse primary hepatocytes with absent or very small LDs, HSD17B13 only partially colocalized with GM4951 in the cytoplasm (Fig. 7a). When primary hepatocytes were loaded with LDs after OA treatment, HSD17B13 and GM4951 almost completely colocalized around LDs to form a "circle-like" distribution (Fig. 7b). Similar results were also observed in AML12 hepatocytes (Supplementary Fig. 7a, b). These data suggest that GM4951 interacts with HSD17B13 and translocates to LDs to regulate lipid metabolism. Although WT GM4951 translocated to LDs in lipid loaded primary hepatocytes (Fig. 7c, d) and AML12 cells (Supplementary Fig. 7c, d), both *Oily* (N86K) and *Carboniferous* (D125G) mutated GM4951 showed dramatically reduced LD translocation (Fig. 7e–h, Supplementary Fig. 7e–h), suggesting that the GTPase activity of GM4951 is important for this translocation. OA induced a strong and continous localization of HSD17B13 around LDs in WT primary hepatocytes (Fig. 7i, j). In *Gm4951*<sup>-/-</sup> primary hepatocytes, the LD localization of HSD17B13 became weak and discontinuous (Fig. 7k, l, Supplementary Fig. 7i). Consistent with this, isolated LDs from *Gm4951*<sup>-/-</sup> primary hepatocytes had decreased protein levels of HSD17B13 by immunoblot (Supplementary Fig. 7j). A similar phenotype was also observed in primary hepatocytes isolated from HFD-fed *Gm4951*<sup>-/-</sup> mice (Fig. 7m, n). In contrast, another LD protein, Perilipin 2 (PLIN2) still maintained LD localization in the absence of GM4951 (Fig. 7o, p). Taken together, these data suggest that GM4951 brings HSD17B13 to LDs in a GTPase activity dependent manner.

**Identification of potential human homologs of mouse GM4951.** Mouse *Gm4951* was not included in the NCBI HomoloGene database. A search for human protein sequences with similarity to mouse GM4951 protein sequence using Basic Local Alignment Search Tool (BLAST) identified two IRGs: immunity related GTPase cinema (IRGC) and immunity related GTPase M (IRGM), with identities of 35% and 38%, respectively (Fig. 8a, Supplementary Fig. 5a). IRGC contains an IIGP domain from amino acid 20 to 402, making it similar in size (by amino acid length) to the IIGP domain (aa 34-402) in mouse GM4951. Immunostaining in AML12 cells showed that most EEA1 positive endosomes colocalized with IRGC, and a small portion of IRGC partially colocalized with AIF positive mitochondria (Fig. 8b, c).

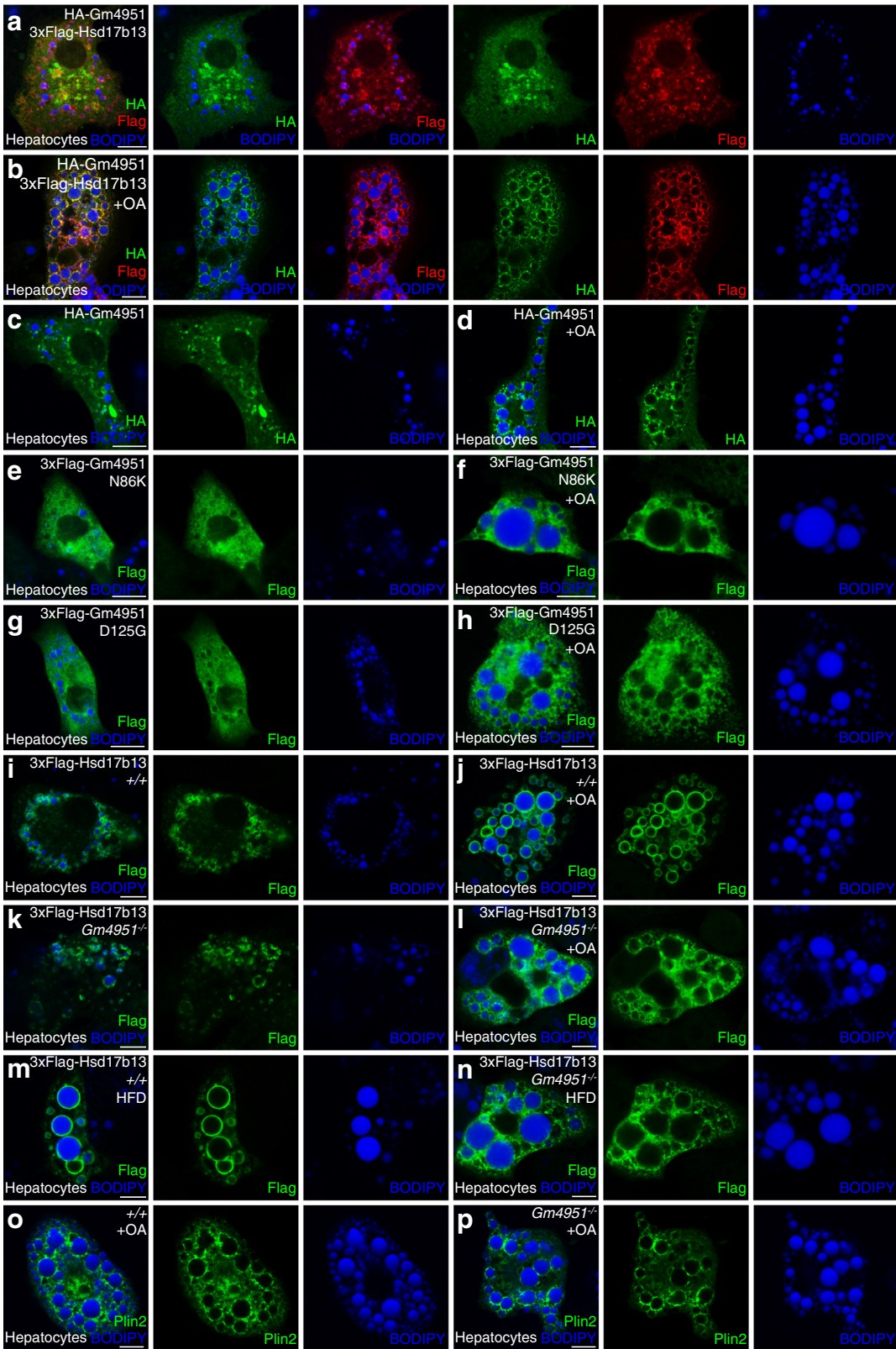

IRGM contains a smaller IIGP domain (aa 2-193) than GM4951. Immunostaining in AML12 cells showed that IRGM exclusively colocalized with AIF positive mitochondria, with no overlap with EEA1 positive endosomes (Fig. 8d, e). It has been reported that variants of IRGM contribute to the development of human NAFLD by altering hepatic lipid metabolism through an effect on autophagy[29]. Thus, we checked the autophagic flux in primary hepatocytes isolated from WT and $Gm4951^{-/-}$ mice. No dramatic difference in LC3BI/II and p62 expression was observed between $Gm4951$ knockout and WT hepatocytes with or without the treatment of the autophagy inhibitor bafilomycin A1 (Supplementary Fig. 8a). No consistent difference in autophagic flux was observed between WT and $Gm4951^{-/-}$ liver lysates (Supplementary Fig. 8b). Interaction between human HSD17B13 and

**Fig. 7 GM4951 targets HSD17B13 to LDs through GTPase activity. a, b** Mouse primary hepatocytes expressing HA-tagged Gm4951 and 3xFlag-tagged Hsd17b13 were immunostained with HA antibody (green), Flag antibody (red), and BODIPY (blue) to visualize LDs. 0.5 mM oleic acid for 24 h was used in +OA groups. **c, d** Mouse primary hepatocytes expressing HA-tagged Gm4951 without or with OA treatment were immunostained with HA antibody (green) and BODIPY (blue) to visualize LDs. **e–h** Mouse primary hepatocytes expressing 3xFlag-tagged Gm4951 mutants without or with OA treatment were immunostained with Flag antibody (green) and BODIPY (blue) to visualize LDs. **i–l** Hepatocytes isolated from WT or Gm4951−/− mice expressing 3xFlag-tagged Hsd17b13 without or with OA treatment were immunostained with Flag antibody (green) and BODIPY (blue) to visualize LDs. Hepatocytes were from mice maintained on chow diet (**a–l**). **m, n** Hepatocytes isolated from 4 week HFD-fed WT or Gm4951−/− mice expressing 3xFlag-tagged Hsd17b13 were immunostained with Flag antibody (green) and BODIPY (blue) to visualize LDs. **o, p** Primary hepatocytes isolated from +/+ or Gm4951−/− mice maintained on chow diet were treated with OA and immunostained with Plin2 antibody (green) and BODIPY (blue) to visualize LDs. (All scale bars: 10 μm.) Data are representative of two independent experiments.

IRGC, but not IRGM (v1, longer isoform; v2, shorter isoform), was observed in 293T cells (Fig. 8f). In cultured primary human hepatocytes, overexpression of GM4951 or IRGC decreased the level of triglycerides after OA treatment, while no reduction of triglycerides was observed in IRGM overexpressing hepatocytes after OA treatment (Fig. 8g, h). *IRGC* was shown to be expressed almost exclusively in human testis (Human Protein Atlas; proteinatlas.org)[30]. Quantification of *IRGC* mRNA by real-time PCR revealed a low level of *IRGC* in primary human hepatocytes, and this level was elevated compared with that detected in primary human pre-adipocytes and other human cell lines (SW1088, astrocytoma cell line; SK-LMS-1, leiomyosarcoma cell line; SW872, liposarcoma cell line; THP-1, monocytic cell line; 293T, embryonic kidney cell line; primary pre-adipocytes; HepG2, hepatocarcinoma cell line) (Fig. 8i). Knockdown of *IRGC* in OA treated primary human hepatocytes decreased the expression of several lipid oxidation genes, including *PPARA*, *ACOT1*, and *ACOX1* (Fig. 8j). All these data favor IRGC as a biologically relevant human homolog of mouse GM4951.

## Discussion

To identify new regulators of NAFLD, we utilized unbiased forward genetic screening and AMM to identify mutations that cause NAFLD in mice sensitized by a HFD for 4 weeks. Two semi-dominant missense alleles of *Gm4951*, named *Oily* and *Carboniferous*, were detected in this screen. GM4951 deficient mice had dramatically increased hepatic lipid accumulation without a concomitant increase in body weight on a HFD. *Gm4951* knockout liver also showed decreased expression of lipid oxidation genes. Moreover, HFD caused a reduction in the level of *Gm4951* in WT mice, which may promote the development of NAFLD. Mechanistically, we have shown that GM4951 interacted with LD protein HSD17B13, and translocated to LDs in response to lipids. The *Oily* and *Carboniferous* mutations abolished the intrinsic GTPase activity of GM4951 and blocked its translocation to LDs. HSD17B13 translocation to LDs required GM4951.

We did not observe differences in body weight, fat weight, fasting glucose, or insulin between *Gm4951*−/− and WT mice even after HFD feeding for 24 weeks (Fig. 3k–n). *Gm4951*−/− mice did not show any differences from WT mice in body weight, fat weight, and glucose metabolism on chow diet either (Fig. 3c–f). These data suggest that the NAFLD caused by GM4951 deficiency is a liver intrinsic effect that reflects the role of GM4951 in hepatocytes, consistent with the predominantly hepatic expression of GM4951. Besides the liver, low but detectable levels of GM4951 existed in the spleen, thymus, lung, eWAT, and iWAT (Fig. 4c). Similar to other IRG genes, GM4951 is an interferon inducible GTPase. Thus, it would be interesting to check whether GM4951 is directly involved in the resistance to intracellular pathogens in the liver or other GM4951 expressing tissues. As the most common cause of chronic liver disease, NAFLD often coexists with liver infections such as hepatitis C. We showed that interferon directly induced the expression of

*Gm4951* in primary hepatocytes (Fig. 4i, j). Further experiments are needed to check whether hepatosteatosis is affected by liver infections through the interferon-GM4951 axis.

GM4951 was not classified as an IRG protein initially, probably due to the variation in the IRG domain with other IRG family proteins. Nevertheless, GM4951 shares 77% amino acid sequence identity with IRGA6, a well characterized IRG protein. Indeed, we observed many similar features between these two proteins. IRGA6 has been shown as a GTPase to undergo nucleotide-dependent oligomerization and cooperatively hydrolyze GTP, acting as its own GTPase-activating protein (GAP)[31–33]. Here we have shown that GM4951 is a bona fide GTPase (Fig. 4k). The N86K substitution of *Oily* is predicted to occur near the GTP-binding pocket and the D125G substitution of *Carboniferous* affects a residue within the predicted binding pocket; both N86 and D125 are conserved in IRGA6 and the *Oily* and *Carboniferous* mutations disrupted the ability of GM4951 to hydrolyze GTP, but not to homodimerize. In response to parasitic infection, IRGA6 translocates from the ER to the parasitophorous vacuole in a mechanism that is dependent on its binding to GTP, a subsequent conformational change, and interactions with mouse IRGM1, IRGM2, and IRGM3[34–36]. Notably, we also identified IRGM1 as a GM4951 interacting protein by mass spectrometry (Supplementary Data 1), raising the possibility that GM4951 may also depend on IRGM1 for translocation to LDs. Similar to IRGA6, GTP binding may affect the conformation of GM4951 and interactions with partner proteins such as IRGM1. Interestingly, *Irgm1*-deficient macrophages exhibited increased glycolysis and accumulation of long chain acylcarnitines[37]. Future study is needed to test the relationship between IRGM1 and GM4951 in lipid metabolism. Our finding of *Oily* and *Carboniferous* mutants highlights the importance of GTPase activity of GM4951 in the regulation of hepatic lipid. Lipid loading appears to positively influence the LD translocation of GM4951, but also to decrease transcription of *Gm4951*.

We utilized Flag tag knockin mice to search for endogenous GM4951 interacting proteins under physiological conditions by mass spectrometry. Although HSD17B13 has been identified as a NAFLD-associated protein both in mice and humans, the reported phenotypes in human patients and different mouse models are not consistent. Two loss-of-function alleles of HSD17B13 in humans are associated with reduced risk of chronic liver disease and protection from progression to steatohepatitis from steatosis[16,38]. Another human study reported that HSD17B13 plays an important role in promoting NAFLD through its retinol dehydrogenase activity[18]. *Hsd17b13* knockout mice were first reported to develop severe hepatic steatosis with aging (9-month-old)[17], similar to that reported here in *Gm4951* knockout mice, which developed significant hepatic steatosis on chow diet in 6-month-old animals (Fig. 3a). *Hsd17b13* knockout mice exhibited NAFLD despite normal body weight, overall body fat content, and glucose metabolism[17], which is also consistent with the key features of NAFLD development in *Gm4951*

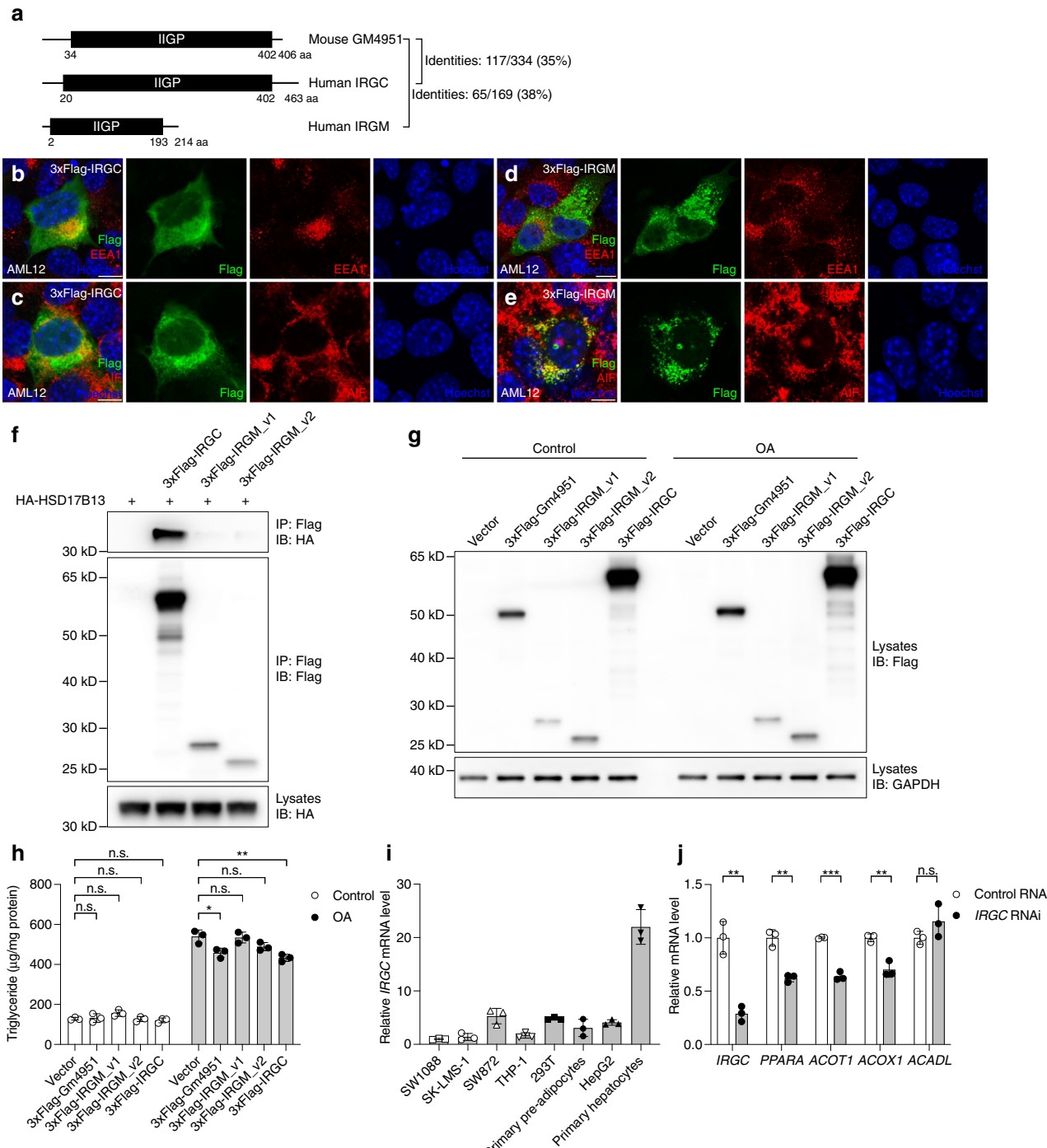

**Fig. 8 Identification of potential human homologs of mouse GM4951. a** Protein domain diagrams of mouse GM4951 and human IRGC and IRGM. Locations of IIGP domains and amino acid identities across the full length of the proteins are indicated. **b–e** AML12 cells expressing 3xFlag-tagged IRGC or IRGM were immunostained with Flag antibody (green), Hoechst 33342 (blue) to visualize nuclei, and organelle markers (red): AIF (mitochondria) and EEA1 (endosome). (Scale bar: 10 μm.) **f** Immunoblots of immunoprecipitates (top and middle) or lysates (bottom) of 293T cells expressing HA-tagged human HSD17B13 and 3xFlag-tagged human IRGC, IRGM_v1, and IRGM_v2. **g, h** Immunoblots of lysates (**g**) or triglyceride level in lysates (*n* = 3 cultures per group) (**h**) of primary human hepatocytes expressing different IRG genes with or without 0.5 mM OA treatment for 24 h. **i** Relative mRNA level of *IRGC* from different human cell lines or human primary cells. Levels were normalized to *ACTB* mRNA and then to levels in SW1088 cells (*n* = 3 cultures per group). **j** Relative mRNA level of *IRGC* and several lipid oxidation genes (*PPARA*, *ACOT1*, *ACOX1*, and *ACADL*) from primary human hepatocytes transfected with control siRNA or *IRGC* siRNA with 0.5 mM OA treatment for 24 h (*n* = 3 cultures per group). Levels were normalized to *ACTB* mRNA and then to levels in control RNAi. Data points represent independent cultures. Data are presented as means ± SD. *P* values were determined by one-way ANOVA with Tukey's multiple comparison test (**h**) or two-tailed Student's *t* test (**j**). *P* values are denoted by *$P < 0.05$; **$P < 0.01$; ***$P < 0.001$; ns, not significant with $P > 0.05$. The exact *P* values of statistically significant groups are: 0.0158 (**h**, Vector vs 3xFlag-Gm4951 OA); 0.0026 (**h**, Vector vs 3xFlag-IRGC); 0.0019 (**j**, *IRGC*); 0.0015 (**j**, *PPARA*); 0.0001 (**j**, *ACOT1*); 0.0031 (**j**, *ACOX1*). Data are representative of two independent experiments (**b–j**).

knockout mice (Fig. 3c–g). These data support the physiological significance of our finding that GM4951 associates with HSD17B13. A recent paper reported a different observation that *Hsd17b13* knockout mice did not show differences from WT mice in hepatic steatosis, liver injury, fibrosis, and inflammation under several fatty liver-inducing dietary conditions[39]. It has also been reported that overexpression of HSD17B13 in mouse liver increases lipogenesis and triglyceride content accompanying higher liver SREBP-1 maturation and fatty acid synthase protein expression[15]. Further study is needed to explain the discrepancy of these results and the exact role of HSD17B13 in NAFLD in mice and humans.

There are 20 IRG genes in mice and two IRG genes, encoding IRGM and IRGC, in humans[7,8]. Human IRGM variants are associated with NAFLD and IRGM directly affects autophagy flux to modulate autophagic degradation of LDs (lipophagy)[29]. While this manuscript was in preparation, it was reported that GM4951 (also known as IFGGA2) induces lipophagy to prevent hepatic lipid accumulation[40]. IRGM has been proposed to be the human orthologue of GM4951[40]. However, we compared features of protein domain structures and subcellular localizations between mouse GM4951 and human IRGC and IRGM, which suggested a different conclusion. Compared to GM4951, IRGC has a similar length of IIGP domain and a similar subcellular localization, whereas IRGM has a much shorter IIGP domain and a distinct mitochondria exclusive subcellular localization (Supplementary Fig. 5a, Fig. 8a–e). The *Carboniferous* mutation site (D125) is conserved among GM4951, IRGA6, IRGC, and IRGM, while the *Oily* mutation site (N86) is only conserved among GM4951, IRGA6, and IRGC (Supplementary Fig. 5a). More importantly, overexpression of either IRGC or GM4951 reduced triglyceride accumulation in primary human hepatocytes, while overexpression of IRGM had no such effect. The interaction between GM4951/IRGC and HSD17B13 is also conserved in human cells. Consistent with the supportive role of GM4951 in lipid oxidation (Fig. 5), loss of IRGC in primary human hepatocytes resulted in decreased expression of several lipid oxidation genes (Fig. 8j). IRGM participates in the clearance of intracellular pathogens through autophagy, while IRGC is not known to be involved in immunity[7]. The function of IRGC is largely unknown. Although detectable in a low level in primary human hepatocytes, *IRGC* is reportedly expressed almost exclusively in human testis (Human Protein Atlas; proteinatlas.org)[30], which is different from the predominantly hepatic expression of GM4951 in mice. The reduced number of IRG proteins in humans relative to mice and low expression of IRGC in human liver might explain why IRGC has not yet been implicated in NAFLD in humans. Further study is needed to determine whether human IRGC is associated with NAFLD though a mechanism similar to that used by mouse GM4951.

An array of hepatocyte nuclear factor (HNF) binding elements constitutes a liver-specific promoter upstream of *Irga6* transcripts[41]. We found a similar array of HNF elements upstream of exon 1 of *Gm4951*. Similar to *Irga6*[41], the promoter region of *Gm4951* also contains several IFN-stimulated response elements (ISRE) and gamma-activated sequences (GAS). These data could explain the predominantly hepatic and IFNγ inducible expression of *Gm4951*. Long-term HFD decreased liver *Gm4951* transcript expression to a level below that in livers from *Gm4951^Oily/+*  or  *Gm4951^Carboniferous/+*  mice, which develop hepatic steatosis (Fig. 5h, i). This suggests that *Gm4951* down-regulation may be an important step during the development of diet induced NAFLD. Studying the transcriptional regulation of *Gm4951* might lay the groundwork for the future development of approaches to transcriptionally activate the human *Gm4951* ortholog to combat NAFLD.

## Methods

**Mice**. C57BL/6 J mice (stock# 000664) were purchased from The Jackson Laboratory. Eight- to ten-week old C57BL/6 J males were mutagenized with ENU[42]. Mutagenized G0 males were bred to C57BL/6 J females, and G1 male progeny were crossed to C57BL/6 J females to produce G2 mice. G2 females were backcrossed to their G1 fathers to yield G3 mice, which were screened for mutations that modify the NAFLD phenotype. Three- to six-month old G3 mice were first weighed then put on a high-fat diet (HFD, 60 kcal% fat, Research Diets). After two weeks of HFD feeding, G3 mice were weighed and then fasted overnight. The next morning, fasting blood samples were collected, and HFD was then reinitiated. Two hours later, blood samples were collected. HFD was continued for two weeks. After a total of four weeks of HFD feeding, G3 mice were weighed and euthanized in the morning with liver samples and terminal blood samples collected. Whole-exome sequencing and mapping were performed as described[6]. The *Oily* (C57BL/6J-Gm4951^Oily) and *Carboniferous* (C57BL/6J-Gm4951^Carboniferous) strains were generated by ENU mutagenesis and are described at http://mutagenetix. utsouthwestern.edu. *Gm4951* knockout (Gm4951^−/−) mice were generated in our laboratory using the CRISPR/Cas9 system[43] with the *Gm4951* (5′-ATCGAAGT-GATGATCTCATG-3′) small base-pairing guide RNA. 3xFlag-*Gm4951* knockin mice were generated in our laboratory, using CRISPR/Cas9-mediated gene replacement with the following sgRNA and oligo template: sgRNA, 5′-CCTTTATTGC TTGAAGTGAT-3′, oligo, 5′-TGATTCCTTTCTCACATTCCTTTAAGTATTCA ATAAAGCTGGAGGACAAATCTTGGTCCTCACTCCTACGTGAAGAGAACA GTTGACCCActtgtcatcgtcatccttgtaatcgatatcatgatctttataatcaccgtcatggtctttgtag tccatCACTTCAAGCAATAAAGGCACTGGAAGAAAGAAGGG-3′.

All mice were fed standard chow diet (2016 Teklad Global 16% Protein Rodent Diet) except mice with diet-induced obesity, which were fed with a HFD. All mice were housed at room temperature (18–23 °C) with 40–60% humidity in a 12 h light/12 h dark cycle. Mice were maintained at the University of Texas Southwestern Medical Center and studies were performed in accordance with institutionally approved protocols. All experiments in this study were approved by the University of Texas Southwestern Medical Center Institutional Animal Care and Use Committee.

**Blood/serum chemistries, ELISA, liver lipids, liver lipid oxidation, hepatic triglyceride secretion, hydrodynamic tail vein injection, and body composition measurement**. Mice were fasted overnight (6:00PM − 8:00AM) for fasting blood sample collections. After overnight fasting, mice were refed for 2 h with their diet prior to fasting and refeeding blood samples were collected. Blood glucose was tested with the AlphaTRAK glucometer and test strips. ELISA kits were used to measure insulin (Crystal Chem) in the serum according to the manufacturer's instructions. ALT was measured with ALT Activity Assay Kit from Sigma-Aldrich. Triglycerides were measured with Infinity Triglycerides Liquid Stable Reagent (Thermo Fisher Scientific). FFA was measured with HR Series NEFA-HR (2) kit (Wako). Cholesterol was measured with Infinity Cholesterol Liquid Stable Reagent (Thermo Fisher Scientific).

To measure liver triglycerides, frozen liver samples were weighed and homogenized in homogenization buffer (10 mM Tris-Cl pH 7.4, 0.9% wt/vol NaCl, and 0.1% vol/vol Triton X-100) using a Tissue Lyser (QIAGEN). Liver homogenates were diluted and incubated at 37 °C with 1% wt/vol deoxycholate. Liver triglycerides were measured with Triglycerides Liquid Stable Reagent (Thermo Fisher Scientific) similarly as serum triglyceride measurement. Liver triglycerides were normalized with total protein (mg) or liver sample weight (g). Liver cholesterol was measured with Cholesterol Liquid Stable Reagent (Thermo Fisher Scientific) similarly as serum cholesterol measurement. Frozen liver samples were weighed and homogenized in 1% weight/vol Triton X-100 chloroform solution and dried to extract lipid for measuring FFA with Free Fatty Acid Quantitation Kit (Sigma-Aldrich).

Liver lipid oxidation was tested with a protocol modified from[44]. Fresh liver pieces (~200 mg) were weighed and rinsed in PBS, then homogenized in chilled STE buffer (10 mM Tris-Cl pH 7.4, 1 mM EDTA, and 0.25 M Sucrose). Liver lysates were centrifuged and supernatants were mixed with oxidation buffer [10 mM Tris-Cl pH 8.0, 5 mM KH$_2$PO$_4$, 100 mM Sucrose, 0.2 mM EDTA, 80 mM KCl, 1 mM MgCl$_2$, 2 mM L-Carnitine, 0.1 mM Malate, 0.05 mM CoA, 2 mM ATP, 1 mM DTT, 1/10 Palmitate solution (7 % BSA, 5 mM Palmitate, 0.01 μCi/μl $^{14}$C-palmitate)] for 30 min incubation at 37 °C. The reaction mix was transferred to the CO$_2$ collection tube for scintillation couting of released CO$_2$, and the remaining solution was centrifuged to scintillation count supernatant for acid soluble acyl-CoAs.

To measure hepatic triglyceride secretion, mice were intravenously injected with 100 μl of 10% (vol/vol) Triton WR-1399 (Sigma-Aldrich) in PBS (Gibco). Blood was collected at 0 h, 1 h, 2 h, and 4 h. Serum was separated and assayed for triglycerides with Infinity Triglycerides Liquid Stable Reagent (Thermo Fisher Scientific). Hydrodynamic tail vein injection was performed on six weeks old male C57BL/6 J mice weighed 18 − 20 g. Mice were put on a heat pad and anaesthetized with isoflurane. Each mice were i.v. injected with 50 μg plasmid DNA in 2 ml saline within 4–8 s at a constant rate. After recovery, all mice were put on a HFD for two weeks before euthanization to check liver triglycerides and gene expression. MRI of live mice was measured by EchoMRI Body Composition Analyzers with default settings.

**Plasmids**. PCR was carried out using mouse liver cDNA as the template and oligonucleotide primers designed to obtain the coding DNA sequence (CDS) of the mouse *Gm4951* (NM_001033767.3) and *Hsd17b13* (NM_001163486.1). Human HepG2 cell cDNA was used to obtain the CDS of the human *IRGM* (v1, NM_001346557.2; v2, NM_001145805.2;) and *IRGC* (NM_019612.4). These genes were cloned into the HA-tagged or 3xFlag-tagged pcDNA vector for transient expression, HA-tagged or 3xFlag-tagged pBOB vector for packaging lentivirus, or GST-tagged pGEX4T1 vector for protein expression in *E. coli*. The *Gm4951* mutants were generated with PCR mutagenesis. All constructs were verified by sequencing.

**Cell culture, transfection, infection, mouse primary hepatocytes isolation, LD isolation, human primary hepatocytes culture, and knockdown by siRNA**. The 293T cells (CRL-3216) were purchased from American Type Culture Collection (ATCC) and grown in culture medium [DMEM (Gibco), 10% (vol/vol) FBS (ATCC), Pen-Strep antibiotics (Gibco)] at 37 °C with 5% CO$_2$. Transfection of plasmids was carried out using Lipofectamine 2000 (Life Technologies) according to the manufacturer's instructions. Cells were harvested between 36 and 48 h post-transfection. The AML12 cells (CRL-2254) were purchased from ATCC and grown in culture medium [DMEM:F12 (ATCC), 10% (vol/vol) FBS (ATCC), 10 μg/ml insulin (Sigma-Aldrich), 5.5 μg/ml transferrin (Sigma-Aldrich), 5 ng/ml selenium (Sigma-Aldrich), 40 ng/ml dexamethasone (Sigma-Aldrich), Pen-Strep antibiotics (Gibco)] at 37 °C with 5% CO$_2$. Infections of AML12 cells were carried out using a 3rd generation lentiviral system packaged in 293T cells[45]. The SW1088 (HTB-12) and SW872 (HTB-92) cells were purchased from ATCC and grown in culture medium [L-15 (ATCC), 10% (vol/vol) FBS (ATCC), Pen-Strep antibiotics (Gibco)] at 37 °C with 100% air. The SK-LMS-1 (HTB-88) and HepG2 (HB-8065) cells were purchased from American Type Culture Collection (ATCC) and grown in culture medium [MEM (ATCC), 10% (vol/vol) FBS (ATCC), Pen-Strep antibiotics (Gibco)] at 37 °C with 5% CO$_2$. The THP-1 (TIB-202) cells were purchased from American Type Culture Collection (ATCC) and grown in culture medium [RPMI-1640 (Gibco), 10% (vol/vol) FBS (ATCC), 0.05 mM 2-mercaptoethanol (Sigma-Aldrich), Pen-Strep antibiotics (Gibco)] at 37 °C with 5% CO$_2$. The primary pre-adipocytes (PCS-210-010) were purchased from American Type Culture Collection (ATCC) and grown in complete growth medium (Fibroblast Growth Kit-Low Serum, ATCC PCS-201-041) at 37 °C with 5% CO$_2$. Mouse primary hepatocytes were isolated with a two-step collagenase perfusion technique[46]. Anesthetized mice were first perfused with medium I [10 mM HEPES (pH 7.4), 1x PBS, 0.05% (wt/vol) KCl, 5 mM Glucose, 0.2 mM EDTA] and then perfused with medium II [10 mM HEPES (pH7.4), 1x PBS, 5 mM Glucose, 1 mM CaCl$_2$] with 1 mg/ml Collagenase Type 4 (CLS-4, Worthington). Isolated hepatocytes were cultured in attachment medium [William's E (Gibco), 10% FBS (ATCC), 1% (vol/vol) non-essential amino acids (Gibco), 1x GlutaMAX (Gibco), Pen-Strep antibiotics (Gibco)]. 6 h after attachment, the medium was replaced with culture medium [William's E (Gibco), 1% (vol/vol) nonessential amino acids (Gibco), 1x GlutaMax (Gibco), Pen-Strep antibiotics (Gibco)]. Infections of primary hepatocytes were carried out using a 3rd generation lentiviral system packaged in 293T cells[45]. To induce the formation of LDs, 500 μM oleic acid (Sigma-Aldrich) was added into the medium for 24 h. Cells were incubated with indicated concentration of IFNγ (R&D Systems) in culture medium for 4 h before harvest. To inhibit autophagy, 100 nM bafilomycin A1 (Sigma-Aldrich) was added into hepatocyte culture for 2 h before harvest. LDs were isolated from hepatocytes treated with 0.5 mM OA for 24 h with Lipid Droplet Isolation Kit (Cell Biolabs). Primary human hepatocytes were purchased from Zen-Bio and grown in hepatocyte plating medium for lentivirus infection overnight. The next morning, medium was changed to hepatocyte maintenance medium with or without 0.5 mM OA for 24 h. Triglycerides were measured using a method similar to the liver triglyceride protocol described above. For the knockdown assay, control siRNA or Accell SMART pool IRGC siRNA (Dharmacon) was added into the culture medium of primary human hepatocytes to a final concentration of 1 μM, and inbucated for 72 h before harvesting the cells. 0.5 mM OA was added into these cells 24 h before harvesting.

**Immunohistochemistry, immunostaining, and image quantification**. Samples for routine histology and special stains were harvested from anesthetized mice and fixed according to standard procedures[47,48] with modifications for tissue size and stains. Samples for routine H&E staining were fixed for 48 h in 10% (vol/vol) neutral-buffered formalin and samples for Oil Red O (ORO) staining were fixed in methanol-free 4% (vol/vol) paraformaldehyde for 48 h before equilibration in 18% (wt/vol) sucrose. Subsequent paraffin processing and embedding (H&E) and cryoembedding (ORO) were carried out, and sections were cut on a rotary microtome and cryostat, respectively. The resulting sections were stained for routine histopathological evaluation by regressive H&E on a Sakura Finetek DRS-601 robotic staining system using Leica SelecTech reagents (hematoxylin 560 and alcoholic eosin Y 515). ORO, PSR, and αSMA staining were performed manually according to established protocols. For immunostaining, AML12 cells or primary hepatocytes cultured in slide chambers were fixed in freshly made 4% (vol/vol) formaldehyde in PBS buffer at room temperature for 15 min, then washed three times with PBS, blocked with blocking buffer [(1xPBS, 5% vol/vol normal goat serum, 0.3% vol/vol Triton X-100), Cell Signaling Technology] for 1 h at room temperature. After blocking, cells were incubated with primary antibody diluted in antibody dilution buffer [(1xPBS, 1% BSA, 0.3% Triton X-100), Cell Signaling Technology] overnight at 4 °C, then washed with PBS and incubated with secondary antibody diluted in antibody dilution buffer for 1 h at room temperature, and finally mounted in mounting medium (Life Technologies). The following primary antibodies were used in this study: mouse anti-Flag (M2, Sigma-Aldrich, 1:500), rabbit anti-HA [C29F4, Cell Signaling Technology (CST), 1:800), anti-AIF (D39D2, CST, 1:400), anti-PDI (C81H6, CST, 1:100), anti-EEA1 (C45B10, CST, 1:200), anti-RCAS1 (D2B6N, CST, 1:200), anti-αSMA (D4K9N, CST, 1:400), and anti-PLIN2 (EPR3713, Abcam, 1:200). The following secondary antibodies were used in the study: Alexa Fluor 488 Goat anti-Mouse IgG (H + L) (115-545-166, Jackson ImmunoResearch, 1:500), Alexa Fluor 488 Goat anti-Rabbit IgG (H + L) (115-545-144, Jackson ImmunoResearch, 1:500), Rhodamine Red-X Goat anti-Mouse IgG (H + L) (115-295-146, Jackson ImmunoResearch, 1:500), and Rhodamine Red-X Goat anti-Rabbit IgG (H + L) (111-295-144, Jackson ImmunoResearch, 1:500). BODIPY (Life Technologies) was used to stain LDs and Hoechst (CST) was used to stain nuclei. SignalStain Boost IHC Detection Reagent (HRP, Rabbit) (#8114, CST, 50 ul) was used in immunohistochemical assays. Immunostaining images were taken with a Zeiss LSM 880 inverted confocal using ZEN software. Quantification of histological stains in liver sections was performed using the threshold function in ImageJ 1.53k. The original RGB image was split into red, green, and blue channels, and the green channel was transformed into an 8-bit grayscale file to use for threshold adjustment and measurement. The threshold for each kind of staining was adjusted to match the actual stain, but was kept the same among different images for the same kind of staining for comparison. Stained area was defined as threshold positive area and expressed as percentage of the total area of an image. The NAS score[26] was defined as the unweighted sum of the scores for steatosis (0–3), lobular inflammation (0–3), and ballooning (0–2); thus ranging from 0 to 8. A NAS score less than 3 means no NASH, higher than 4 means NASH, and scores of 3 or 4 mean indeterminate.

**Sample preparation, immunoprecipitation, mass spectrometric analysis, GST pull-down, western blot analysis, and GTPase assay**. To check protein level in whole cell lysates, cells were harvested in 1× NuPAGE LDS sample buffer (Life Technologies) with 2.5% (vol/vol) 2-mercaptoethanol (Sigma-Aldrich). For immunoprecipitation, cells were lysed in Nonidet P-40 lysis buffer [50 mM Tris-Cl, pH 8.0, 100 mM NaCl, 10 mM sodium fluoride, 1 mM sodium vanadate, 1% (vol/vol) Nonidet P-40, 10% (vol/vol) glycerol, 1.5 mM EDTA, and Protease Inhibitor Mixture] for 30 min at 4 °C. After centrifugation, lysates were incubated with Flag antibody-conjugated beads (M2, Sigma-Aldrich) for 2 h at 4 °C. Beads were washed three times with 1 mL of Nonidet P-40 lysis buffer and then eluted with 3xFlag peptides (Sigma-Aldrich) for 30 min at 4 °C. The procedure of immunoprecipitation for mass spectrometric analysis was the same except the whole liver was snap frozen in liquid nitrogen and lysed with homogenizer. Mass spectrometric analysis was performed as previously described[45]. GST fusion proteins were produced in *Escherichia coli* BL21 and purified with glutathione agarose beads (GE Healthcare). 3xFlag-tagged protein was purified with anti-Flag beads from 293T cells. GST fusion protein-loaded beads were incubated with eluted 3xFlag proteins in GST pull-down buffer [20 mM Tris-Cl, pH 8.0, 200 mM NaCl, 1 mM EDTA, 0.5% (vol/vol) Nonidet P-40, and PMSF] at 4 °C for 2 h. The beads were washed three times with GST pull-down buffer, followed by immunoblot analysis. For immunoblot analysis, samples were resolved by NuPAGE 4–12% (wt/vol) Bis-Tris gels (Thermo Fisher Scientific), transferred to nitrocellulose membranes (Bio-Rad), blotted with the primary antibody at 4 °C overnight and the secondary antibody for 1 h at room temperature, and then visualized by chemiluminescent substrate (Thermo Fisher Scientific). The following primary antibodies were used in this study: mouse anti-HA (HA-7, Sigma-Aldrich, 1:5000), anti-Flag (M2, Sigma-Aldrich, 1:5000), rabbit anti-Gapdh (D16H11, CST, 1:2000), anti-β-tubulin (9F3, CST, 1:2000), anti-p62 (D6M5X, CST, 1:1000), anti-LC3B (D3U4C, CST, 1:1000), and anti-HSD17B13 (gift from Dr. Helen Hobbs, 1:2000). The following secondary antibodies were used in this study: Goat anti-Mouse IgG (H + L) HRP (115-035-146, Jackson ImmunoResearch, 1:5000), Goat anti-Mouse IgG (light chain specific) HRP (115-035-174, Jackson ImmunoResearch, 1:5000), and Goat anti-Rabbit IgG (H + L) HRP (111-035-144, Jackson ImmunoResearch, 1:5000). GTPase assay was performed with purified WT or mutant GM4951 proteins using GTPase-Glo Assay kit (Promega) for 2 h at room temperature.

**RNA isolation, reverse transcription, and quanitative PCR**. Tissue samples or cells were lysed in TRIzol (Invitrogen) for RNA isolation following a standard protocol, and 1 μg of RNA was used for reverse transcription by SuperScript III First-Strand Synthesis SuperMix (Life Technologies). Quanitative PCR was performed with ABI StepOnePlus with Powerup SYBR Green Master Mix (Life Technologies). The $2^{-\Delta\Delta Ct}$ method was used for relative quantification. The primer information is listed in Supplementary Table 1.

**Reporting summary**. Further information on research design is available in the Nature Research Reporting Summary linked to this article.

# Data availability
The mass spectrometry proteomics data generated in this study have been deposited in the MassIVE repository with the accession code MSV000089462. All other data are

available in the main article or the Supplementary Information files. Source data are provided with this paper.

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

## Acknowledgements

We thank the Histo Pathology, Mass Spectrometry, Live Cell Imaging, and Metabolic Phenotyping Cores of the University of Texas Southwestern Medical Center for providing excellent services. We thank Dr. Helen Hobbs at the University of Texas Southwestern Medical Center for providing HSD17B13 antibodies and helpful discussions, Dr. Jay Horton at the University of Texas Southwestern Medical Center for helping with lipid oxidation experiments. This work was supported by National Institutes of Health grants R00 DK115766 (Z.Z.), R01 DK130959 (Z.Z.), R01 AI125581 (B.B.), and U19 AI100627 (B.B.), and by the Lyda Hill foundation (B.B.). This work was also supported in part by a sponsored research agreement from Pfizer, Inc.

## Author contributions

Conceptualization, Z.Z., B.B; Data curation, Z.Z., B.B; Formal analysis, Z.Z., B.B.; Funding acquisition, Z.Z., B.B.; Investigation, Z.Z., Y.X., S.R., L.Y., J.S.; Methodology, Z.Z., B.B.; Project administration, Z.Z., B.B.; Resources, Z.Z., X.L., M.T., K.K., S.L., B.B.; Software, Z.Z., B.B.; Supervision, Z.Z., B.B.; Validation, Z.Z., B.B.; Visualization, Z.Z.; Writing- original draft, Z.Z.; Writing- review and editing, Z.Z., E.M.Y.M., B.B.

## Competing interests

The authors declare no competing interests.
