## [Peer Review File · Nature Communications]

Reviewers' Comments:

Reviewer #1:

Remarks to the Author:

Zhang et al identify GM4951/Ifgga2 as a gene/protein that regulates hepatic lipid metabolism and suppress the development of NAFLD. They show that it localizes to lipid droplets in hepatocytes, and establish a partial mechanism through which the protein functions - by regulating translocation of HSD17B13 to LDs. This manuscript is important in establishing a novel factor and regulatory pathway that controls NAFLD. The manuscript follows a logical progression and the writing is excellent. The experimental data fully support the conclusions and are high quality, performed rigorously, and subjected to appropriate statistical analyses. This manuscript will make an important addition to the literature.

The manuscript could be improved by addressing the following:

1. The authors refer several time to the Immunity Related GTPase (IRG), IIGP1. This is the name under which that protein/gene was originally described. However, the nomenclature of the IRG protein family was systematized several years ago, and IIGP1 is now more commonly referred to as IRGA6. It is suggested that the authors change the text to refer to IIGP1 as 'IRGA6/IIGP1' or 'IRGA6'.

There are about 20 proteins in the mouse IRG family. As IIGP1 is the focus of much text, is the sequence of IRGA6 the closest to that of GM4951 among all of the mouse IRG proteins? Because GM4951 was not classified as an IRG protein initially, does it vary in some way with the other IRG proteins? Clarifying these points would be helpful.

Fig. S4 shows an alignment of a small region of GM4951 and IRGA6. However, it seems of interest to display how the full protein sequences compare, as well as a comparison to human IRG proteins. It is recommended to aid the readers that in the supplemental figures, the authors include a full alignment of mouse GM4951, mouse IRGA6, perhaps other mouse IRGs, and human IRGM and/or IRGC with the GTPase, dimerization, and other regions of interest identified.

2. The authors refer to an interferon inducible GTPase (IIGP) domain (e.g. lines 154, 349). This is not referenced, and it is not clear to what they are referring. Is this the consensus GTPase domain among the proteins in the IRG protein family? The IRG GTPase domain is thought to be very similar to the GTPase domains found in many other GTP-binding proteins. It is recommended that the authors elaborate on this point and provide appropriate references.

3. The authors reach the conclusion that human IRGC rather than human IRGM is the appropriate homologue to mouse GM4951. It is notable that human IRGC is thought to be expressed almost exclusively in the testis. The authors should consider how this shapes their hypothesis in the text.

4. Another mouse IRG protein, IRGM1, has been shown to affect lipid metabolism in macrophages [Schmidt et al, J Biol Chem (2017) 292(11): 4651-4662.] This should be referenced and appropriately considered in the text.

5. In Fig 7K, the authors reach the conclusion that in GM4951^{-/-} cells, 'the majority of HSD17B13 did not reside around LDs' (lines 270-1). This is an interpretation of their IF images. The staining pattern of HSD17B13 is indeed qualitatively altered; nevertheless, another interpretation is that the protein appears to remain associated with LDs, but have a more uneven or discontinuous pattern. If the authors agree, perhaps the text could be altered appropriately and/or available biochemical data could be presented to clarify.

Reviewer #2:

Remarks to the Author:

In this study, Zhang Z et al., have performed ENU-mutagenesis-based in vivo forward genetic screen to identify genes important for NAFLD pathogenesis. Through this screen, they found 2 hot

spot mutations in the same gene, Gm4951, associated with the elevation of liver triglyceride levels. Subsequently, they generated Gm4951 knockout mice and showed that Gm4951 deficiency exacerbated NAFLD, exhibited by the increase in liver triglyceride accumulation and liver injury. Gm4951 was shown to be mainly expressed in the liver, especially hepatocytes and mechanistic studies demonstrated that Gm4951 was required to bring HSD17B13 to lipid droplet via GTPase activity in hepatocytes.

NAFLD is the global health issue affecting 30% of the people over the world. There is no effective therapy for this disease so it is no doubt desired to understand its molecular pathogenesis and identify therapeutic targets. This group is the front-runner of ENU-mutagenesis-based in vivo forward genetic screen and has successfully identified hundreds of genes important for a variety of phenotypes. In this paper, authors elegantly discovered the novel gene important for the development of NAFLD through their sophisticated in vivo screening tool and also provided the concrete evidence of its involvement in the NAFLD pathogenesis by generating KO mice. The study design is appropriate, the study finding is novel and the manuscript is overall well-written. So I believe this paper is potentially interesting to the people in this field and may give new insight to them.

On the other hand, the biggest criticism and limitation of this paper is the unknown clinical relevance of their finding to human NAFLD. As authors also mentioned in the paper, there is no human homologs of Gm4951. So their findings can't be directly applied to clinical setting. This limitation is quite significant and more rigorous work to identify its human counterpart can't be exempted before the further consideration.

Another major criticism for this paper is the insufficiency of mechanistic insight. Although they showed the role of Gm4951 in the delivery of HSD17B13 to LD in hepatocytes in vitro, it's in vivo significance in the NAFLD pathogenesis remain unclarified. This part needs substantial improvement.

All other comments and questions are shown below.

Major

- 1) To prove the clinical significance of their findings in human NAFLD, validation of candidate genes of human counterpart of Gm4951 is mandatory. Comprehensive functional studies for IRGC are needed including but not limited to KO phenotypes in mice and human cells. Ideally, the clinical significance of human counterpart should be evaluated in the human sample.
- 2) The role of Gm4951 in lipid oxidation is not sufficiently investigated. It is totally unclear how subtle changes of lipid oxidation-related gene expressions really affect in vivo phenotype. The fatty acid oxidation was really impaired in the KO animal? Is the same effect observed in the hepatocytes in vitro? Is it just the consequence or causal factor? If this is the causal factor of the NAFLD exacerbation, how does Gm4951 affect lipid oxidation pathway? Gm4951KO mice spontaneously develop NAFLD at 24 weeks of age with chow diet, so how about the expressions of lipid metabolism-related genes in Gm4951KO mice w/o HFD at later stage than 4 weeks?
- 3) In vivo phenotypes of NAFLD mice need to be more carefully examined. Most important aspect of NAFLD is liver fibrosis which is strongly linked to patient outcome. This aspect needs to be investigated including Sirius red staining of liver tissue, a variety of fibrosis-related gene expressions, activation of HSCs, etc.
- 4) Lipid profiling of KO mice is also insufficient. What about the FFA, cholesterol in the liver? What about lipid profile in the serum?
- 5) The biological meaning of the role of Gm4951 to bring HSD17B13 to LD is unclear. What is the consequence when HSD17B13 can't go to LD in vitro? Is this process required for fatty acid oxidation? Is this universal phenomenon irrespective of type of fatty acid? Are the same phenomenon also observed in the Gm4951 KO mice with HFD feeding?
- 6) It is unclear whether the NAFLD phenotype of Gm4951 KO mice was caused by its knockout in hepatocytes in vivo. Authors need to extensively check the lipid metabolism in adipose tissues in KO mice.
- 7) As therapeutic point of view, can you show any evidence that induction or upregulation of Gm4951 improve NAFLD?
- 8) What is the mechanism of Gm4951 downregulation in the HFD-fed mice?
- 9) Does IFN treatment upregulate Gm4951 in the liver in mice?
- 10) Is there any alteration of autophagy in the liver of Gm4951 KO mice?

Minor.

- 1) forward genetic screen part was fully undescribed. More detailed experimental methodology has to be provided.
- 2) Isolation method of a variety of NPC in the liver should be described.
- 3) What was the difference between WT and REF in Fig 1
- 4) Most of the labels of graphs are too small to read.
- 5) Data should be provided as S.D. but not S.E.M.

Reviewer #3:

Remarks to the Author:

The authors of this manuscript use an elegant forward genetic screen to identify two mutations of the Gm4951 gene in hepatocytes that regulate hepatic triglyceride accumulation in response to a high fat diet. The mechanistic role of Gm4951 was elucidated by generation of an in vivo tagged protein that GM4951 regulated the translocation of HSD17B13 to lipid droplets thereby reducing triglyceride accumulation. If HSD17B13 does not translocate to the lipid droplet, this results in increased triglyceride accumulation. The presumed mechanism for triglyceride accumulation is based upon several studies which pointed to regulation of PPAR alpha and oxidation genes. GM4951 was found to express an interferon inducible GTPase. Interestingly, hepatic triglyceride accumulation occurred in absence of any effects on whole body adiposity or insulin-glucose metabolism

several questions are raised

- 1) Gene expression studies performed by the authors are used to suggest that fat oxidation is the underlying mechanism by which GM4951 regulates triglyceride accumulation. The authors should perform actual fat oxidation studies to confirm a role for fat oxidation in the effects of GM4951 to regulate triglyceride accumulation
- 2) A possible explanation for reduced hepatic levels is increased free fatty acids availability to activate tPPAR alpha. One possible explanation that would funnel fatty acids to oxidation would be increased expression of adipose tissue triglyceride lipase or the proteins localization to the lipid droplet which should be investigated. Could the association of GM4951 and HSD17B13 regulate ATGL association with the lipid droplet
While the authors suggest that hepatic

Reviewer #1 (Remarks to the Author):

Zhang et al identify GM4951/Ifgga2 as a gene/protein that regulates hepatic lipid metabolism and suppress the development of NAFLD. They show that it localizes to lipid droplets in hepatocytes, and establish a partial mechanism through which the protein functions - by regulating translocation of HSD17B13 to LDs. This manuscript is important in establishing a novel factor and regulatory pathway that controls NAFLD. The manuscript follows a logical progression and the writing is excellent. The experimental data fully support the conclusions and are high quality, performed rigorously, and subjected to appropriate statistical analyses. This manuscript will make an important addition to the literature.

The manuscript could be improved by addressing the following:

1. The authors refer several time to the Immunity Related GTPase (IRG), IIGP1. This is the name under which that protein/gene was originally described. However, the nomenclature of the IRG protein family was systematized several years ago, and IIGP1 is now more commonly referred to as IRGA6. It is suggested that the authors change the text to refer to IIGP1 as 'IRGA6/IIGP1' or 'IRGA6'.

We thank Reviewer #1 for the good suggestion. All IIGP1 has been changed to IRGA6.

There are about 20 proteins in the mouse IRG family. As IIGP1 is the focus of much text, is the sequence of IRGA6 the closest to that of GM4951 among all of the mouse IRG proteins? Because GM4951 was not classified as an IRG protein initially, does it vary in some way with the other IRG proteins? Clarifying these points would be helpful.

Yes, the sequence of IRGA6 the closest to that of GM4951 among all of the mouse IRG protein, and GM4951 varies in some way with the other IRG proteins. We have clarified these points in Results and Discussion.

Fig. S4 shows an alignment of a small region of GM4951 and IRGA6. However, it seems of interest to display how the full protein sequences compare, as well as a comparison to human IRG proteins. It is recommended to aid the readers that in the supplemental figures, the authors include a full alignment of mouse GM4951, mouse IRGA6, perhaps other mouse IRGs, and human IRGM and/or IRGC with the GTPase, dimerization, and other regions of interest identified.

Yes, we now added a new Fig. S5a to compare the full protein sequence of GM4951, IRGA6, human IRGC, and human IRGM. Residues involved in dimer formation or nucleotide interaction were marked according to the structure of IRGA6.

2. The authors refer to an interferon inducible GTPase (IIGP) domain (e.g. lines 154, 349). This is not referenced, and it is not clear to what they are referring. Is this the consensus GTPase domain among the proteins in the IRG protein family? The IRG GTPase domain is thought to be very similar to the GTPase domains found in many other GTP-binding proteins. It is recommended that the authors elaborate on this point and provide appropriate references.

The IIGP domain in GM4951 was identified by protein sequence analysis using SMART (<http://smart.embl-heidelberg.de>). IIGP is a Pfam domain (PF05049), which does not have a single consensus but was built on 6 seed sequences (TGTP1, IRGA6, IFI47, IRGM1, IGTP, IRGM2) for alignment. We have added the reference for Pfam, the Pfam accession number, and more information about this Pfam domain in the revision.

3. The authors reach the conclusion that human IRGC rather than human IRGM is the appropriate homologue to mouse GM4951. It is notable that human IRGC is thought to be expressed almost exclusively in the testis. The authors should consider how this shapes their hypothesis in the text.

IRGC is expressed almost exclusively in human testis according to the Human Protein Atlas (proteintlas.org), which is different from the predominantly hepatic expression of GM4951 in mice. The reduced number of IRG proteins in humans (2) relative to mice (20) and low expression of IRGC in human liver might explain why IRGC has not yet been implicated in NAFLD in humans. It is possible that IRGC only expresses in hepatocytes under certain physiological or pathological conditions. We have added this information in the discussion. We also added new data to support the conclusion that human IRGC is a more appropriate homologue to mouse GM4951, including: 1) Overexpression of IRGC rather than IRGM decreased triglyceride accumulation in primary human hepatocytes (Fig. S6f,g); 2) Human IRGC rather than IRGM strongly interacted with human HSD17B13 in human cells (Fig. S7a).

4. Another mouse IRG protein, IRGM1, has been shown to affect lipid metabolism in macrophages [Schmidt et al, J Biol Chem (2017) 292(11): 4651–4662.] This should be referenced and appropriately considered in the text.

We thank Reviewer #1 for pointing out this important paper. We also identified IRGM1 as a GM4951 interacting protein by mass spectrometry. It is possible that GM4951 works together with IRGM1 for LD translocation and lipid metabolism regulation. Now we have cited this paper and added this content to the Discussion.

5. In Fig 7K, the authors reach the conclusion that in GM4951^{-/-} cells, ‘the majority of HSD17B13 did not reside around LDs’ (lines 270-1). This is an interpretation of their IF images. The staining pattern of HSD17B13 is indeed qualitatively altered; nevertheless, another interpretation is that the protein appears to remain associated with LDs, but have a more uneven or discontinuous pattern. If the authors agree, perhaps the text could be altered appropriately and/or available biochemical data could be presented to clarify.

Yes, we agree with this and revised the text appropriately.

Reviewer #2 (Remarks to the Author):

In this study, Zhang Z et al., have performed ENU-mutagenesis-based in vivo forward genetic screen to identify genes important for NAFLD pathogenesis. Through this screen, they found 2 hot spot mutations in the same gene, Gm4951, associated with the elevation of liver triglyceride levels. Subsequently, they generated Gm4951 knockout mice and showed that Gm4951 deficiency exacerbated NAFLD, exhibited by the increase in liver triglyceride accumulation and liver injury. Gm4951 was shown to be mainly expressed in the liver, especially hepatocytes and mechanistic studies demonstrated that Gm4951 was required to bring HSD17B13 to lipid droplet via GTPase activity in hepatocytes.

NAFLD is the global health issue affecting 30% of the people over the world. There is no effective therapy for this disease so it is no doubt desired to understand its molecular pathogenesis and identify therapeutic targets. This group is the front-runner of ENU-mutagenesis-based in vivo forward genetic screen and has successfully identified hundreds of genes important for a variety of phenotypes. In this paper, authors elegantly discovered the novel gene important for the development of NAFLD through their sophisticated

in vivo screening tool and also provided the concrete evidence of its involvement in the NAFLD pathogenesis by generating KO mice. The study design is appropriate, the study finding is novel and the manuscript is overall well-written. So I believe this paper is potentially interesting to the people in this field and may give new insight to them.

On the other hand, the biggest criticism and limitation of this paper is the unknown clinical relevance of their finding to human NAFLD. As authors also mentioned in the paper, there is no human homologs of Gm4951. So their findings can't be directly applied to clinical setting. This limitation is quite significant and more rigorous work to identify its human counterpart can't be exempted before the further consideration.

Another major criticism for this paper is the insufficiency of mechanistic insight. Although they showed the role of Gm4951 in the delivery of HSD17B13 to LD in hepatocytes in vitro, it's in vivo significance in the NAFLD pathogenesis remain unclarified. This part needs substantial improvement.

All other comments and questions are shown below.

Major

1) To prove the clinical significance of their findings in human NAFLD, validation of candidate genes of human counterpart of Gm4951 is mandatory. Comprehensive functional studies for IRGC are needed including but not limited to KO phenotypes in mice and human cells. Ideally, the clinical significance of human counterpart should be evaluated in the human sample.

We totally understand the reviewer's request to identify the human counterpart of GM4951. The following approaches were tried:

1. We overexpressed GM4951, IRGC, and IRGM (two isoforms) in human primary hepatocytes with lentivirus, and measured triglyceride accumulation induced by oleic acid (OA). As shown in Fig. S6f,g, overexpression of GM4951 or IRGC, but not IRGM, reduced the level of triglyceride in OA treated human primary hepatocytes.

2. We tested the interaction between human HSD17B13 and IRGC or IRGM (two isoforms) and only found a strong interaction between human HSD17B13 and IRGC (Fig. S7a).

3. We deleted *IRGC* in human HepG2 cells by CRISPR/Cas9 gene targeting. We obtained 4 *IRGC* KO clones. After treatment with 0.5 mM OA for 24 h, we observed similar levels of triglycerides in *IRGC*-deficient and -sufficient cells (Fig. R1, this negative data was not included in the revision). As one of the most commonly used hepatocellular carcinoma cells, we realized that HepG2 might not be a good model to study human NAFLD.

4. To check the clinical significance of human *IRGC* or *IRGM* polymorphisms, we collaborated with Dr. Helen Hobbs (UT Southwestern), who has a large collection (Dallas Heart Study, 9502 individuals) of human genetic data regarding disorders of lipid metabolism. No association between *IRGC* or *IRGM* polymorphisms and hepatic triglycerides was found in data from the Dallas Heart Study.

Figure for Reviewer (Fig. R) 1. The effect to KO IRGC in HepG2 cells. Triglyceride of *pLentiCRISPR* control and *IRGC* KO HepG2 cells 24 h after 0.5 mM OA treatment.

Based on our current data, we still favor IRGC rather than IRGM as human counterpart of GM4951, based on the following evidence:

1. Compared to GM4951, IRGC has a similar length of IIGP domain and a similar subcellular localization, whereas IRGM has a much shorter IIGP domain and a distinct mitochondria exclusive subcellular localization (Figs. S5a and S6).
2. Overexpression of GM4951 or IRGC decreased triglyceride levels in human primary hepatocytes (Fig. S6f,g). Although we did not see a *IRGC* KO phenotype in HepG2 cells, we consider human primary hepatocytes to be a better model to study NAFLD compared with HepG2. Due to the limited number of days human primary hepatocytes can be cultured *ex vivo*, we could not generate KO clones in these cells with CRISPR.
3. IRGC rather than IRGM interacted with human HSD17B13 (Fig. S7a), suggesting the regulation of HSD17B13 by IRGC might be conserved in humans. As Reviewer #1 also pointed out, IRGC is shown to be expressed almost exclusively in human testis from Human Protein Atlas (proteinatlas.org), which is different from the predominantly hepatic expression of GM4951 in mice. The reduced number of IRG proteins in humans (2) relative to mice (20) and low expression of IRGC in human liver might explain why IRGC has not yet been implicated in NAFLD in humans. It is possible that IRGC only expresses in hepatocytes under certain physiological or pathological conditions, which would be a good research direction for future studies.

2) The role of Gm4951 in lipid oxidation is not sufficiently investigated. It is totally unclear how subtle changes of lipid oxidation-related gene expressions really affect *in vivo* phenotype. The fatty acid oxidation was really impaired in the KO animal? Is the same effect observed in the hepatocytes *in vitro*? Is it just the consequence or causal factor? If this is the causal factor of the NAFLD exacerbation, how does Gm4951 affect lipid oxidation pathway? Gm4951KO mice spontaneously develop NAFLD at 24 weeks of age with chow diet, so how about the expressions of lipid metabolism-related genes in Gm4951KO mice w/o HFD at later stage than 4 weeks?

In the revision, we measured lipid oxidation activity by proteins in liver lysates towards ^{14}C -palmitate *in vitro*. As shown in Fig. 5e,f, *Gm4951* KO livers had significantly reduced lipid oxidation activity. In C57BL6/J mice, four-week HFD feeding increases both liver triglyceride and liver lipid oxidation. Since GM4951 deficiency increases liver triglyceride while decreases lipid oxidation, we believe that decreased lipid oxidation is the causal factor rather than the consequence. We expect the role of GM4951 in lipid oxidation to involve HSD17B13 and their translocation to LDs, but we do not fully understand the molecular and cellular consequences of GM4951-HSD17B13 interaction or translocation to LDs (or their failure). These are questions we are actively investigating; we expect this will be a substantial body of work that will stand alone in a separate publication.

As requested by Reviewer #2, we tested the expression of lipid metabolism-related genes in mice fed a normal chow diet for 24 weeks and saw a similar trend of decreased expression of *Ppara*, *Acox1*, *Acot1*, and *Acadl* lipid oxidation genes in *Gm4951*^{-/-} livers relative to expression in WT livers (Fig. 5c).

3) *In vivo* phenotypes of NAFLD mice need to be more carefully examined. Most important aspect of NAFLD is liver fibrosis which is strongly linked to patient outcome. This aspect needs to be investigated including Sirius red staining of liver tissue, a variety of fibrosis-related gene expressions, activation of HSCs, etc.

We thank Reviewer #2 for making this important point. In the revision, we examined liver fibrosis in GM4951 deficient mice in several ways:

1. Picrosirius red (PSR) staining of liver sections showed increased collagen in HFD-fed *Gm4951*^{-/-} mice (Fig. S4e,f).
2. The mRNA levels of several genes known to be upregulated in fibrotic livers (*Colla1*, *Mmp9*, *Timp1*) were increased in livers from HFD-fed *Gm4951*^{-/-} mice (Fig. S4i).
3. α -smooth muscle actin (α -SMA) staining of liver sections showed increased α -SMA-positive myofibroblasts, which suggests the activation of hepatic stellate cells (HSCs) (Fig. S4g,h).
- 5) The biological meaning of the role of Gm4951 to bring HSD17B13 to LD is unclear. What is the consequence when HSD17B13 can't go to LD in vitro? Is this process required for fatty acid oxidation? Is this universal phenomenon irrespective of type of fatty acid? Are the same phenomenon also observed in the Gm4951 KO mice with HFD feeding?

We isolated primary hepatocytes from WT or *Gm4951*^{-/-} mice fed with HFD for 4 weeks. Immunostaining showed that Flag-tagged HSD17B13 localized to LDs in primary hepatocytes from the WT mice. However, HSD17B13 failed to localize to LDs in hepatocytes from *Gm4951*^{-/-} mice, similar to the effect observed when hepatocytes were treated with oleic acid in vitro. As mentioned in the response to comment #2, we do not understand the molecular and cellular consequences of HSD17B13 translocation to LDs. This is a question we are actively investigating; we expect this will be a substantial body of work that will stand alone in a separate publication.

- 6) It is unclear whether the NAFLD phenotype of Gm4951 KO mice was caused by its knockout in hepatocytes in vivo. Authors need to extensively check the lipid metabolism in adipose tissues in KO mice.

To address this question, we checked the mRNA levels of lipid metabolism genes (*Acaca*, *Pnpla2*, *Dgat1*, *Fasn*), adipokines (*Fabp4*, *Adipoq*, *Leptin*), and key transcription factors (*Fabp4*, *Cebpa*) in the inguinal white adipose tissue (iWAT) of 4 week HFD-fed *Gm4951* KO mice. The expression levels of all tested genes were similar between WT and *Gm4951*^{-/-} iWAT (Fig. S3a).

- 7) As therapeutic point of view, can you show any evidence that induction or upregulation of Gm4951 improve NAFLD?

We delivered 3xFlag-Gm4951 expression plasmid or empty vector into WT mice through hydrodynamic tail vein injection, which is known to induce relatively specific gene expression in liver. After two weeks on HFD, mice overexpressing 3xFlag-Gm4951 in the liver (Fig. 4e) had reduced levels of liver triglycerides compared to mice injected with the empty vector (Fig. 4f). This finding suggests that upregulation of GM4951 can reduce NAFLD.

- 8) What is the mechanism of Gm4951 downregulation in the HFD-fed mice?

We don't know the exact mechanism of Gm4951 downregulation in the HFD-fed mice. We have added the following text to the Discussion to address this comment: "*An array of hepatocyte nuclear factor (HNF) binding elements constitutes a liver-specific promoter upstream of Irga6 transcripts. We found a similar array of HNF elements upstream of exon 1 of Gm4951. Similar to Irga6, the promoter region of Gm4951 also contains several IFN-stimulated response elements (ISRE) and gamma-activated sequences*

(*GAS*).” We believe these elements are the first thing to check in HFD-fed mice to study the downregulation of *Gm4951* in the future.

9) Does IFN treatment upregulate Gm4951 in the liver in mice?

Our goal in this manuscript was to understand how GM4951 influences liver triglycerides and we feel it is beyond the scope of the current paper to investigate immune-mediated regulation and function of GM4951. While we have shown that IFN γ treatment can stimulate Gm4951 expression in cultured cells, understanding whether and under which conditions IFN γ affects liver Gm4951 expression *in vivo* will require a more in-depth future study.

10) Is there any alteration of autophagy in the liver of Gm4951 KO mice?

We analyzed LC3B and p62 expression as a readout for autophagic flux in the livers of WT and *Gm4951* KO mice and did not observe consistent differences between genotypes (Fig. S6e).

Minor.

1) forward genetic screen part was fully undescribed. More detailed experimental methodology has to be provided.

The detailed experimental methodology regarding the forward genetic screen has been added in the “Mice” section of Methods.

2) Isolation method of a variety of NPC in the liver should be described.

The distribution of GM4951 in different cell types in the liver was obtained from another publication (Azimifar et al, Cell Metabolism, 2014), thus we did not put the method in our paper. We have made this more clear and the citation has been included both in the main text and figure legend.

3) What was the difference between WT and REF in Fig 1

REF refers to G3 mice homozygous for the reference (WT) allele of the gene in question (*Gm4951*); G3 mice carry other ENU-induced mutations. WT mice are C57BL6/J mice age-matched to G3 mice. We clarified this in the figure legend.

4) Most of the labels of graphs are too small to read.

In the revision, we enlarged the labels of graphs whenever possible.

5) Data should be provided as S.D. but not S.E.M.

All data are now presented as S.D.

Reviewer #3 (Remarks to the Author):

The authors of this manuscript use an elegant forward genetic screen to identify two mutations of the the *Gm4951* gene in hepatocytes that regulate hepatic triglyceride accumulation in response to a high fat diet.

The mechanistic role of Gm4951 was elucidated by generation of an in vivo tagged protein that GM4951 regulated the translocation of HSD17B13 to lipid droplets thereby reducing triglyceride accumulation. If HSD17B13 does not translocate to the lipid droplet, this results in increased triglyceride accumulation. The presumed mechanism for triglyceride accumulation is based upon several studies which pointed to regulation of PPAR alpha and oxidation genes. GM4951 was found to express an interferon inducible GTPase. Interestingly, hepatic triglyceride accumulation occurred in absence of any effects on whole body adiposity or insulin-glucose metabolism

several questions are raised

1) Gene expression studies performed by the authors are used to suggest that fat oxidation is the underlying mechanism by which GM4951 regulates triglyceride accumulation. The authors should perform actual fat oxidation studies to confirm a role for fat oxidation in the effects of GM4951 to regulate triglyceride accumulation

In this revised manuscript, we measured lipid oxidation activity by proteins in liver lysates towards ¹⁴C-palmitate *in vitro*. As shown in Fig. 5e,f, *Gm4951* KO livers had significantly reduced lipid oxidation activity.

2) A possible explanation for reduced hepatic levels is increased free fatty acids availability to activate tPPAR alpha. One possible explanation that would funnel fatty acids to oxidation would be increased expression of adipose tissue triglyceride lipase or the proteins localization to the lipid droplet which should be investigated. Could the association of GM4951 and HSD17B13 regulate ATGL association with the lipid droplet

While the authors suggest that hepatic

Figure for Reviewer (Fig. R) 2. The expression and localization of ATGL in WT and *Gm4951* KO liver. a, Relative mRNA level of *Pnpla2* (*ATGL*) in the liver from WT or *Gm4951* KO mice fed on chow diet or HFD. **b,** Immunoblots of liver lysates from WT or *Gm4951* KO mice fed on chow diet or HFD. **c-f,** Hepatocytes isolated from WT or *Gm4951* KO mice without or with OA treatment were immunostained with ATGL antibody (green) and BODIPY (blue) to visualize LDs.

We thank Reviewer #3 for pointing out this interesting hypothesis. We performed experiments to test this:

1. We checked the mRNA level of *ATGL* (*Pnpla2*) in the livers from chow- and HFD-fed WT and *Gm4951* KO mice (Fig. R2a). No significant change of *ATGL* mRNA level was observed.
2. We checked the protein level of ATGL in the livers from chow -and HFD-fed WT and *Gm4951* KO mice (Fig. R2b). No consistent difference in ATGL expression was observed between WT and *Gm4951* KO mice after HFD feeding.
3. We examined the subcellular localization of ATGL in WT and *Gm4951* KO hepatocytes with or without OA treatment (Fig. R2c-f). ATGL translocated to LDs after OA treatment in *Gm4951* KO hepatocytes similarly to WT hepatocytes.

Based on these data, ATGL is unlikely to be the link between GM4951-HSD17B13 and lipid oxidation.

Reviewers' Comments:

Reviewer #1:

Remarks to the Author:

The authors have done an excellent job addressing my original suggestions to improve the manuscript. I have no remaining concerns. The work described is thorough and high quality. It will make an important addition to the literature.

Reviewer #2:

Remarks to the Author:

Authors performed several additional experiments during the revision and the revised manuscript improved some points. However, based on the authors' responses, I still feel that this manuscript has the significant limitations such as insufficiency of clinical relevance and mechanistic insight. The specific comments are shown below.

1) Authors still fail to sufficiently demonstrate clinical relevance of their findings. If all the findings can't be translated into the biology of human NAFLD (no appropriate counterpart of Gm4951 exist in humans), elegant phenotypic data regarding Gm4951 in the mouse model is just for experimental animal medicine. In this sense, some data of potential human counterpart gene IRGC is promising so that they shouldn't be in the supplement. I strongly suggest to generate Fig.8 that include all the data related to IRGC including S6a,b,c,f,g, S7a. I would also like to see the role of IRGC in lipid oxidation in human hepatocytes in vitro. In the response letter, they showed the negative data of IRGC KO in HepG2 cells and excused by saying that HepG2 is not a good model. It is disappointing because knockdown experiment in human primary hepatocytes could be easily performed. Loss-of-Function phenotype is more important than Gain-of-Function considering the KO phenotype of Gm4951.

The explanation that IRGC is expressed exclusively in human testis but might be expressed in hepatocyte in certain condition is not justified unless otherwise such condition is indeed shown. It is not difficult to check the expression levels of IRGC in the liver of human NAFLD using publicly-available microarray or RNA-seq datasets even if you may not have your own dataset. Even some single cell RNA-seq data is available. At least some clue that IRGC is expressed in liver or hepatocytes under such pathological condition is desired to propose IRGC as a relevant human counterpart. SNP data from the Dallas Heart Study should be also included in the manuscript.

2) Authors failed to demonstrate the role of Gm4951-Hsd17B13 interaction. Unless their role become clear, I can't judge the biological importance in Figure 6 and 7. Only the binding of these 2 molecules does not mean much. So I thus feel that mechanistic insight is still insufficient.

3) Histological phenotypes of NAFLD/NASH were still underrepresented. All HE, Oil-red O, and Sirius red staining are essential in order to evaluate the murine phenotypes of NAFLD/NASH appropriately. They are usually also quantified. Therefore, these data should be provided not in supplement but in the main figures. They also include not only on HFD but also Chow diet with or w/o Gm4951. Authors described "the development of steatohepatitis" in line 123 page 3 but it can't be justified unless liver inflammation and hepatocyte ballooning was exhibited by HE staining together with lipid deposition by Oil-red O. NAS score also should be considered to evaluate the activity of NASH to this end.

Minor)

1) Why the units of liver triglyceride are different between Fig 2c and Fig3a,h. They should be unified.

Reviewer #4:

Remarks to the Author:

This is an interesting study with meticulous methodology and overall valid findings. It was a pleasure to read.

We find that the authors appropriately addressed the comments of original reviewer #3. We do have a few additional comments, which we believe can be addressed easily.

1. We believe that the conclusion (as stated in the paper title) that GM4951 loss leads to NAFLD THROUGH association with HSD17B13 is somewhat of an overstatement. The authors show convincingly that GM4951 loss leads to NAFLD, and show that GM4951 likely interacts with HSD17B13. They have not shown that the latter leads to the former (i.e. the word "through" in the title). We suggest the authors consider changing the title of the paper and toning it down.

2. The issue of the human homolog for GM4951 was raised by other reviewers. The authors suggest IRGC protein as a potential human homolog of the mouse protein, and show an interaction in vitro in an artificial co-expression system. However, since IRGC is expressed almost exclusively in human testis and HSD17B13 is expressed in hepatocytes, this interaction is unlikely to be relevant in vivo. The authors raise an interesting suggestion that "It is possible that IRGC only expresses in hepatocytes under certain physiological or pathological conditions." They can easily verify or disprove this hypothesis, at least preliminarily, by exploring publicly available datasets from databases (i.e. GEO) of diseased human livers. If no evidence for hepatic IRGC expression is found, it is very unlikely to be the relevant human homolog they were seeking and the discussion should be changed accordingly.

3. Figure 7: To demonstrate that GM4951 affects HSD17b13 localization, the authors provide immunofluorescence data (Fig. 7 and S8). However, they only assess localization qualitatively and subjectively, do not show quantitative data, and provide an image of only one cell per condition. Based on the images provided, there is marked overexpression of the target proteins in the selected cells, which may affect localization by itself. We propose the authors quantify localization in multiple cells, with presumably varying levels of expression, and possibly provide additional example. A definitive experiment the authors may consider is isolating lipid droplets from the co-transfected OA treated cells and demonstrate the presence or absence of HSD17B13 on these isolated droplets in the context of GM4951 expression.

4. Figure 4C: The figure caption states "Quantification of bands in the lower panel are shown as ratio of intensity of Gm4951 to Gapdh." But no graph is shown.

5. Figure S4: The PSR and aSMA staining data should be quantified and presented as such.

6. We feel that the new data on GM4951 overexpression in mice (lines 158-162) belongs more with the previous paragraph on p. 3 (after the effects of knockout) and less in this paragraph which deals with cellular localization. We suggest moving but leave it to the authors' decision.

7. The authors used a tag antibody to detect HSD17B13 constructs, with the exception of the IB data in figure S7B, which appears to use a home-made antibody. Do they have validation data for that antibody? HSD17B13 is highly homologous with HSD17B11 and they likely share antigenic regions.

8. Typos:

a. Figure S5: line 899: "mosue" instead of "mouse"

b. Page 10, line 456: "inbucation" instead of "incubation"

c. Page 11, line 495: "500 mM" instead of "500 μ M"

d. Page 11, line 496: "inbucated" instead of "incubated" and "concertation" instead of "concentration"

Reviewer #1 (Remarks to the Author):

The authors have done an excellent job addressing my original suggestions to improve the manuscript. I have no remaining concerns. The work described is thorough and high quality. It will make an important addition to the literature.

Reviewer #2 (Remarks to the Author):

Authors performed several additional experiments during the revision and the revised manuscript improved some points. However, based on the authors' responses, I still feel that this manuscript has the significant limitations such as insufficiency of clinical relevance and mechanistic insight. The specific comments are shown below.

1) Authors still fail to sufficiently demonstrate clinical relevance of their findings. If all the findings can't be translated into the biology of human NAFLD (no appropriate counterpart of Gm4951 exist in humans), elegant phenotypic data regarding Gm4951 in the mouse model is just for experimental animal medicine. In this sense, some data of potential human counterpart gene IRGC is promising so that they shouldn't be in the supplement. I strongly suggest to generate Fig.8 that include all the data related to IRGC including S6a,b,c,f,g, S7a.

We thank Reviewer #2 for the good suggestion. In the revision, we now generated Fig. 8 to include all the data related to IRGC as the reviewer suggested.

I would also like to see the role of IRGC in lipid oxidation in human hepatocytes in vitro. In the response letter, they showed the negative data of IRGC KO in HepG2 cells and excused by saying that HepG2 is not a good model. It is disappointing because knockdown experiment in human primary hepatocytes could be easily performed. Loss-of-Function phenotype is more important than Gain-of-Function considering the KO phenotype of Gm4951.

We thank Reviewer #2 for the good suggestion. We knocked down *IRGC* using siRNA in human primary hepatocytes, and achieved ~70% reduction in the relative mRNA level of *IRGC* 72h after transfection (Fig. 8j). Among four lipid oxidation genes (*PPARA*, *ACOT1*, *ACOX1*, *ACADL*) tested, *IRGC* RNAi resulted in significantly reduced expression of *PPARA*, *ACOT1*, and *ACOX1*, which is similar to the effect in *Gm4951* KO mouse primary hepatocytes (Fig. 5d). These new data suggest similar roles of GM4951 and IRGC in lipid oxidation in mice and humans, respectively. We also found the expression level of *IRGC* in primary hepatocytes is around 5-fold higher than in HepG2 cells (Fig. 8i), which might explain the lack of effect of *IRGC* KO in HepG2 cells that we observed previously.

The explanation that IRGC is expressed exclusively in human testis but might be expressed in hepatocyte in certain condition is not justified unless otherwise such condition is indeed shown. It is not difficult to check the expression levels of IRGC in the liver of human NAFLD using publicly-available microarray or RNA-seq datasets even if you may not have your own dataset. Even some single cell RNA-seq data is available. At least some clue that IRGC is expressed in liver or hepatocytes under such pathological condition is desired to propose IRGC as a relevant human counterpart. SNP data from the Dallas Heart Study should be also included in the manuscript.

We agree with Reviewer #2 that the suggestion that “*It is possible that IRGC only expresses in hepatocytes under certain physiological or pathological conditions...*” in the previous Discussion was not justified and we deleted it. As Reviewer #2 suggested, we checked several publicly-available microarray and RNA-

seq datasets in GEO which tested liver gene expression profiles from the following patients: obesity (GSE7117), NAFLD (GSE49541), hepatitis B virus (GSE38941), hepatitis C virus (GSE48445, GSE11190, GSE6764, GSE38597), GB virus C (GSE16593), and hepatocellular carcinoma (GSE19665, GSE33006). Although we did not see a consistent change in *IRGC* expression under these conditions, *IRGC* transcripts were clearly detected in these various datasets, confirming its expression in the human liver. We do not have direct access to human samples. As an alternative approach, we collected different human cell lines (SW1088, astrocytoma cell line; SK-LMS-1, leiomyosarcoma cell line; SW872, liposarcoma cell line; THP-1, monocytic cell line; 293T, embryonic kidney cell line; HepG2, hepatocarcinoma cell line), primary pre-adipocytes, and primary hepatocytes and measured the expression of *IRGC* by real-time PCR. Despite the low expression of *IRGC* in these human cells, we observed relatively higher expression of *IRGC* in primary hepatocytes compared to other cells (Fig. 8i). Combined with the *IRGC* RNAi data, these new data support *IRGC* as a relevant human counterpart of GM4951.

Since SNP data from the Dallas Heart Study did not find association between either *IRGC* or *IRGM* and hepatic triglycerides, we do not think it would be informative to include these data in the manuscript.

2) Authors failed to demonstrate the role of Gm4951-Hsd17B13 interaction. Unless their role become clear, I can't judge the biological importance in Figure 6 and 7. Only the binding of these 2 molecules does not mean much. So I thus feel that mechanistic insight is still insufficient.

We understand Reviewer #2's concern regarding the mechanistic function of GM4951-HSD17B13 interaction. In addition to the binding of GM4951 and HSD17B13 in Figures 6 and 7, we also showed that GM4951 is necessary for HSD17B13 localization to the LDs, and moreover that GM4951 localization at LDs requires its GTPase activity. As we presented previously, we expect the role of GM4951 in lipid oxidation to involve HSD17B13 and their translocation to LDs, but we do not fully understand the molecular and cellular consequences of GM4951-HSD17B13 interaction or translocation to LDs (or their failure). These are questions we are actively investigating; we expect this will be a substantial body of work that will stand alone in a separate publication.

3) Histological phenotypes of NAFLD/NASH were still underrepresented. All HE, Oil-red O, and Sirius red staining are essential in order to evaluate the murine phenotypes of NAFLD/NASH appropriately. They are usually also quantified. Therefore, these data should be provided not in supplement but in the main figures. They also include not only on HFD but also Chow diet with or w/o Gm4951. Authors described "the development of steatohepatitis" in line 123 page 3 but it can't be justified unless liver inflammation and hepatocyte ballooning was exhibited by HE staining together with lipid deposition by Oil-red O. NAS score also should be considered to evaluate the activity of NASH to this end.

We thank Reviewer #2 for making this important point. We now put these histological phenotypes in the main Fig. 3p-w. We also used ImageJ to quantify these histological stains (Fig. S4). We double checked line 123, page 3 in the original manuscript and we accurately described "*the development of hepatosteatosis*" rather than "*the development of steatohepatitis*" so we did not change it in the revision. NAS score was added in Fig. S4e.

Minor)

1) Why the units of liver triglyceride are different between Fig 2c and Fig3a,h. They should be unified.

We thank Reviewer #2 for pointing this out. We now fixed these by show the units of liver triglyceride as a unified mg.

Reviewer #4 (Remarks to the Author):

This is an interesting study with meticulous methodology and overall valid findings. It was a pleasure to read. We find that the authors appropriately addressed the comments of original reviewer #3. We do have a few additional comments, which we believe can be addressed easily.

1. We believe that the conclusion (as stated in the paper title) that GM4951 loss leads to NAFLD THROUGH association with HSD17B13 is somewhat of an overstatement. The authors show convincingly that GM4951 loss leads to NAFLD, and show that GM4951 likely interacts with HSD17B13. They have not shown that the latter leads to the former (i.e. the word “through” in the title). We suggest the authors consider changing the title of the paper and toning it down.

We thank Reviewer #4 for making this important point. We agree with it and have revised the title to be “Loss of immunity-related GTPase GM4951 leads to nonalcoholic fatty liver disease without obesity.”

2. The issue of the human homolog for GM4951 was raised by other reviewers. The authors suggest IRGC protein as a potential human homolog of the mouse protein, and show an interaction in vitro in an artificial co-expression system. However, since IRGC is expressed almost exclusively in human testis and HSD17B13 is expressed in hepatocytes, this interaction is unlikely to be relevant in vivo. The authors raise an interesting suggestion that “It is possible that IRGC only expresses in hepatocytes under certain physiological or pathological conditions.” They can easily verify or disprove this hypothesis, at least preliminarily, by exploring publicly available datasets from databases (i.e. GEO) of diseased human livers. If no evidence for hepatic IRGC expression is found, it is very unlikely to be the relevant human homolog they were seeking and the discussion should be changed accordingly.

We thank Reviewer #4 for the good suggestion. As suggested, we checked several publicly-available microarray and RNA-seq datasets in GEO which tested liver gene expression profiles from the following patients: obesity (GSE7117), NAFLD (GSE49541), hepatitis B virus (GSE38941), hepatitis C virus (GSE48445, GSE11190, GSE6764, GSE38597), GB virus C (GSE16593), and hepatocellular carcinoma (GSE19665, GSE33006). Although we did not see a consistent change in *IRGC* expression under these conditions, *IRGC* transcripts were clearly detected in these various datasets, confirming its expression in the human liver. Since we do not have direct access to human samples, as an alternative approach we collected different human cell lines (SW1088, astrocytoma cell line; SK-LMS-1, leiomyosarcoma cell line; SW872, liposarcoma cell line; THP-1, monocytic cell line; 293T, embryonic kidney cell line; HepG2, hepatocarcinoma cell line), primary pre-adipocytes, and primary hepatocytes and measured the expression of *IRGC* by real-time PCR. Despite the low expression of *IRGC* in these human cells, we observed relatively higher expression of *IRGC* in primary hepatocytes compared to other cells (Fig. 8i). Combined with the *IRGC* RNAi data, these new data support *IRGC* as a relevant human counterpart of GM4951. We removed the statement that “*It is possible that IRGC only expresses in hepatocytes under certain physiological or pathological conditions...*” in our revision.

3. Figure 7: To demonstrate that GM4951 affects HSD17b13 localization, the authors provide immunofluorescence data (Fig. 7 and S8). However, they only assess localization qualitatively and subjectively, do not show quantitative data, and provide an image of only one cell per condition. Based on the images provided, there is marked overexpression of the target proteins in the selected cells, which may affect localization by itself. We propose the authors quantify localization in multiple cells, with presumably varying levels of expression, and possibly provide additional example. A definitive experiment the authors may consider is isolating lipid droplets from the co-transfected OA treated cells

and demonstrate the presence or absence of HSD17B13 on these isolated droplets in the context of GM4951 expression.

We thank Reviewer #4 for these suggestions. We now quantify the level of HSD17B13 on LD in WT and *Gm4951* KO hepatocytes from multiple cells with various levels of expression (Fig. S7i). We also performed the suggested experiment to examine levels of 3xFlag-HSD17B13 on isolated LDs from OA treated WT and *Gm4951* KO hepatocytes. Immunoblot analysis clearly showed a reduced level of HSD17B13 on LDs in the absence of GM4951 (Fig. S7j).

4. Figure 4C: The figure caption states “Quantification of bands in the lower panel are shown as ratio of intensity of Gm4951 to Gapdh.” But no graph is shown.

We thank Reviewer #4 for noticing this omission. The graph was accidentally removed during figure preparation, but now it has been added back to Fig. 4c.

5. Figure S4: The PSR and aSMA staining data should be quantified and presented as such.

We thank Reviewer #4 for the good suggestion. We used ImageJ to quantify these histological stains and the data are now shown in Fig. S4.

6. We feel that the new data on GM4951 overexpression in mice (lines 158-162) belongs more with the previous paragraph on p. 3 (after the effects of knockout) and less in this paragraph which deals with cellular localization. We suggest moving but leave it to the authors’ decision.

We agree with Reviewer #4 and reorganized the relevant figures and text as suggested.

7. The authors used a tag antibody to detect HSD17B13 constructs, with the exception of the IB data in figure S7B, which appears to use a home-made antibody. Do they have validation data for that antibody? HSD17B13 is highly homologous with HSD17B11 and they likely share antigenic regions.

We received the HSD17B13 antibody as a gift from Dr. Helen Hobbs. This monoclonal antibody was generated in the Hobbs lab and shows a very specific signal for HSD17B13. We validated the specificity of the antibody in immunoblots of liver lysates from WT and *Hsd17b13* KO mice (from Dr. Helen Hobbs). In the molecular range of 15 kD to 140 kD, only a single HSD17B13 band was observed in WT mice. We do not have specific antigen information, but we believe a HSD17B13 specific region was used to generate this monoclonal antibody.

8. Typos:

- Figure S5: line 899: “mosue” instead of “mouse”
- Page 10, line 456: “inbucation” instead of “incubation”
- Page 11, line 495: “500 mM” instead of “500 mM”
- Page 11, line 496: “inbucated” instead of “incubated” and “concertation” instead of “concentration”

We thank Reviewer #4 again for carefully checking our manuscript. All these typos have been fixed now.

Reviewers' Comments:

Reviewer #2:

Remarks to the Author:

The authors did a great job and answered all my concerns. Now I believe that the revised manuscript contains very solid and clinically relevant findings regarding the novel genes important for NAFLD biology. I don't have any further comment on this manuscript.

Reviewer #4:

Remarks to the Author:

The authors have adequately addressed our concerns, and the manuscript is a pleasure to read.